# A Consistent Lebesgue Measure for Multi-label Learning

## Abstract

Multi-label loss functions are usually either non-convex or discontinuous, which is practically challenging or impossible to optimise directly. Instead, surrogate loss functions can quantify and approximate the quality of a predicted label set. However, their consistency with the desired loss functions is not proven. This issue is further exacerbated by the conflicting nature of multi-label loss functions. To learn from multiple related, yet potentially conflicting multi-label loss functions using a unified representation of a model, we propose a *Consistent Lebesgue Measure-based Multi-label Learner* (CLML). We begin by proving that the optimisation of the Lebesgue measure directly corresponds to the optimisation of multiple multi-label losses, *i.e.*, CLML can achieve theoretical consistency under a Bayes risk framework. Empirical evidence supports our theory by demonstrating that: (1) CLML can consistently achieve a better rank than state-of-the-art methods on a wide range of loss functions and datasets; (2) the primary factor contributing to this performance improvement is the Lebesgue measure design, as CLML optimises a simpler feedforward model without additional label graph or semantic embeddings; and (3) an analysis of the results not only distinguishes CLML's effectiveness but also highlights inconsistencies between the surrogate and the desired loss functions. Code and pre-trained weights for CLML are available at `https://github.com/*redacted*`.

## 1 Introduction

In multi-label data, instances are associated with multiple target labels simultaneously. Multi-label learning is an important paradigm applicable to many real-world domains such as tabulated learning (Yeh et al., 2017; Bai et al., 2021; Hang & Zhang, 2022; Lu et al., 2023), functional genomics (Patel et al., 2022), and computer vision (Wang et al., 2020). Deep learning is responsible for many modern advancements in multi-label learning problems (Zhou et al., 2021; Liu et al., 2023).

However, multi-label learning is usually considered challenging due to its complex label interactions. Label graph embedding is one such approach that superimposes label interactions on a representation, *i.e.*, the weights of a deep learning model. As such, label graph embedding has been the primary concern of works such as Wang et al. (2020), and Yuan et al. (2023). In works such as Yeh et al. (2017); You et al. (2020); Hang & Zhang (2022); Yuan et al. (2023), various transformer and auto-encoder architectures have detected complex forms of feature and label interactions. Generally speaking, deep learning for multi-label learning is dominated by computer vision methods, therefore drawing focus away from tabulated problems, an important area of multi-label learning (Yeh et al., 2017; Bai et al., 2021; Patel et al., 2022; Hang & Zhang, 2022; Lu et al., 2023).

Multi-label learning is challenging due to the complexity of the output space. No existing loss function can quantify the quality of a label set in a universal manner. To exemplify this, consider the following loss functions: hamming loss, one minus the label ranking average precision, and one minus the micro-averaged $F_1$-score. All three loss functions ultimately pertain to multi-label accuracy (Han et al., 2023). However, both the interpretation of quality and the learning behaviour can vary with the loss function selected (Wu & Zhu, 2020; Liu et al., 2021). The situation is worsened by the conflicting behaviour between loss functions (Wu & Zhu, 2020). Further, multi-label loss functions are themselves, typically, *non-convex* and *discontinuous*, which can be either challenging or impossible to optimise directly (Gao & Zhou, 2011). As a result, it is common to back-propagate

on gradients obtained from a manually designed and differentiable surrogate loss function (Rumelhart et al., 1986; Liu et al., 2021). However, both the learning behaviour and the solution itself are prescribed by the gradients of the chosen *surrogate* loss function, which might not correspond to desired behaviour according to the desired loss (Raymond et al., 2023). Finally, multi-label learning with surrogate loss functions is not always *consistent* with what they are designed to approximate (Gao & Zhou, 2011; Liu et al., 2021).

These challenges give rise to several interesting *research questions*. First, how can a model learn directly from non-convex, discontinuous, or even non-differentiable loss functions without surrogates to avoid inconsistency? Second, how can a *unified* representation, *i.e.*, a single model, learn using *multiple* related, yet potentially conflicting loss functions? Third, can such a method achieve theoretical consistency in the context of multi-label learning? Addressing the above questions is paramount to progressing the field of multi-label learning, especially tabulated multi-label learning.

These three important research questions motivate the design of a *Consistent Lebesgue Measure-based Multi-label Learner* (*CLML*), offering *several advantages*. First, CLML learns from multiple related, yet potentially conflicting, loss functions using a *unified* representation on tabulated data. Second, CLML learns to solve the problem *without* the use of a surrogate loss function. Third, our experimental findings demonstrate that CLML consistently achieves a $13.4\%$ to $59.4\%$ *better* critical distance ranking against competitive state-of-the-art methods on a variety of loss functions and datasets. The empirical results are supported by our theoretical foundation that *proves* the consistency of CLML when optimising several multi-label loss functions. The importance of CLML's approach is accentuated by its *simple* representation, validating the importance of a consistent loss function for multi-label learning. Finally, our analysis of the optimisation behaviour suggests that CLML can consistently navigate the desired loss landscape while naturally understanding and accounting for their trade-offs.

The *major contributions* of this work are as follows: (1) a novel approach to achieving tabulated multi-label learning with multiple loss functions; (2) a novel learning objective for several non-convex and discontinuous multi-label loss functions without the use of a surrogate loss function; (3) a proof showing that our method can theoretically achieve consistency; (4) results demonstrating that CLML with a simpler feedforward model representation can consistently achieve a better ranking on a wide variety of loss functions and datasets than state-of-the-art methods; (5) analysis to solidify the importance of consistency in multi-label learning; and (6) an analysis that highlights how CLML can naturally consider the trade-offs between desired multi-label loss functions and the inconsistency between surrogate and desired loss functions.

## 2 METHODOLOGY

### 2.1 NOTATIONS FOR MULTI-LABEL CLASSIFICATION

Multi-label learning is a supervised classification task, where an instance can be associated with multiple class labels simultaneously. Let $\mathcal{X} \in \mathbb{R}^D$, $\mathcal{Y} \in \{0,1\}^K$, and $\Omega \in \mathbb{R}^L$ respectively denote the input, output, and learnable parameter space for $D$ features, $K$ labels, and $L$ parameters. Let $\mathcal{P}$ be a joint probability distribution of samples over $\mathcal{X} \times \mathcal{Y}$. Let $f : \mathbb{R}^D \to \mathbb{R}^K$ represent a deep neural network drawn from $\Omega \in \mathbb{R}^L$, and trained on $N$ samples drawn from $\mathcal{P}$. An input vector $\mathbf{x} \in \mathcal{X}$, where $\mathcal{X} \in \mathbb{R}^D$, can be associated with an output vector that is a subset of $\mathcal{Y} \in \{0,1\}^K$, *i.e.*, $\mathbf{y} = \{y_1, ..., y_K\}$, where $y_l = 1$ if label $l$ is associated with $\mathbf{x}$, and is otherwise zero. We define the input feature and label data as $\mathbf{X} \in \mathcal{X}^N$ and $\mathbf{Y} \in \mathcal{Y}^N$, respectively. Given $\mathbf{x} \in \mathcal{X}$, we denote $p(\mathbf{y}|\mathbf{x})_{\mathbf{y} \in \mathcal{Y}}$ as the conditional probability of $\mathbf{y}$. We additionally define $\kappa$ as the set of all conditional probabilities:

$$\kappa = \{p(\mathbf{y}|\mathbf{x}) : \sum_{\mathbf{y} \in \mathcal{Y}} p(\mathbf{y}|\mathbf{x}) = 1 \wedge p(\mathbf{y}|\mathbf{x}) \geq 0\}. \tag{1}$$

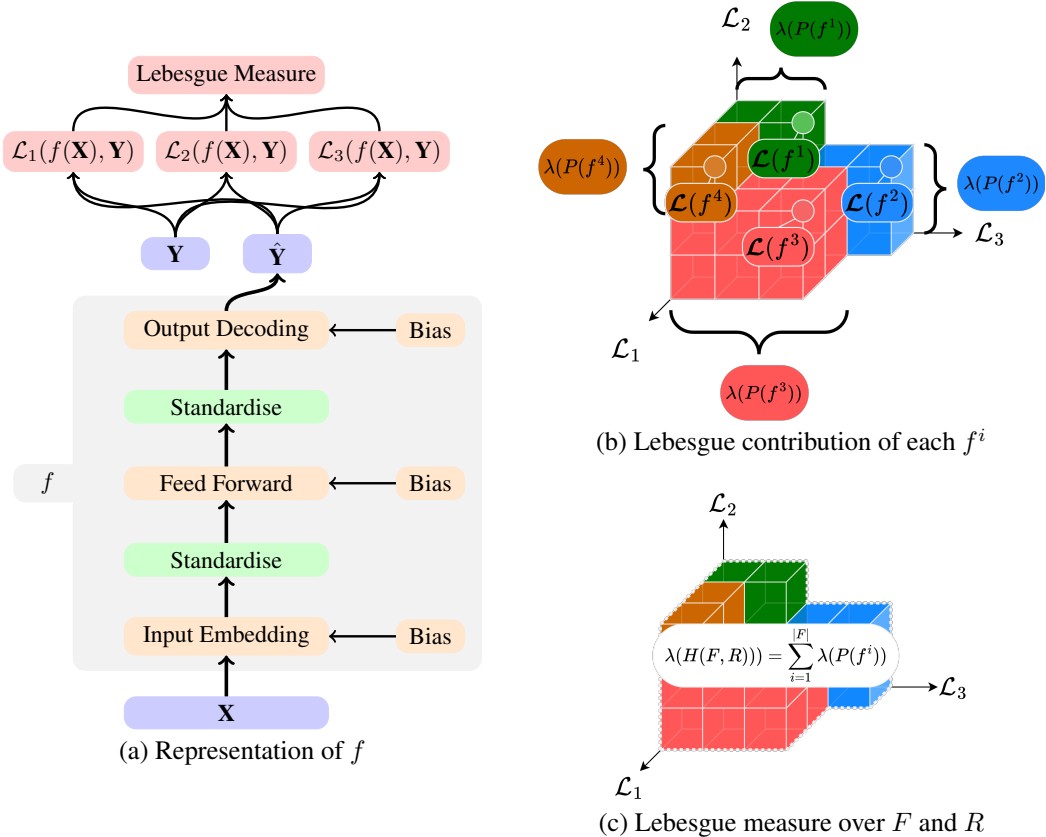

Figure 1: The overall proposed approach of CLML is outlined as follows. (a) illustrates the representation of $f$. (b) illustrates the contribution of each $f^i$ toward the improvement over all three loss functions $\mathcal{L}(f^i) = (\mathcal{L}_1(f^i), \mathcal{L}_2(f^i), \mathcal{L}_3(f^i))$, which is quantified as the non-overlapping volume of space that $\mathcal{L}(f^i)$ uniquely covers over a set of $W$ models $F = \cup_{i=1}^{W}\{f^i\}$, and a reference vector $R = \{1\}^3$. (c) illustrates the overall Lebesgue measure over $F$, which is the aggregate volume of all $f^i \in F$.

and the conditional risk of $f$ given surrogate loss ($\psi$), loss ($\mathcal{L}$), the conditional probability of sample $\mathbf{x}$ and the label set $\mathbf{y}$:

$$\mathcal{L}^c(p(\mathbf{y}|\mathbf{x}), f) = \sum_{\mathbf{y} \in \mathcal{Y}} p(\mathbf{y}|\mathbf{x})\mathcal{L}(f(\mathbf{x}), \mathbf{y})$$

$$\psi^c(p(\mathbf{y}|\mathbf{x}), f) = \sum_{\mathbf{y} \in \mathcal{Y}} p(\mathbf{y}|\mathbf{x})\psi(f(\mathbf{x}), \mathbf{y}).$$

(2)

### 2.2 THE REPRESENTATION OF CLML

Throughout this paper, we use a standard feedforward model to represent $f$, which is illustrated in Figure 1. However, in comparison to a standard feedforward neural network, our model takes matrices as inputs and outputs, rather than individual vectors. This is due to the tabular nature of the data, allowing us to handle all samples simultaneously. First, the encoding layer $\mathbf{E} : \mathbb{R}^{N \times D} \to \mathbb{R}^{N \times C}$ with bias $\mathbf{W}_b^{\mathbf{E}}$, compresses the input signal to $C$ embedding dimensions, where $C << D$. An ablation study of $C$ is given in Section A.3. Note that positional encoding is not required due to the tabulated nature of the data (Vaswani et al., 2017). The compressed input is then row-standardised ($\gamma$) before being passed through a feedforward layer ($\mathbf{L}$) with weights $\mathbf{W}^{\mathbf{L}}$ and bias $\mathbf{W}_b^{\mathbf{L}}$. We repeat standardisation before passing each row to the decoder $\mathbf{D} : \mathbb{R}^{N \times C} \to \mathbb{R}^{N \times K}$ with bias $\mathbf{W}_b^{\mathbf{D}}$. The

full equation for generating the prediction matrix $\hat{\mathbf{Y}}$ is given by:

$$\hat{\mathbf{Y}} = \sigma(\sigma(\gamma(\sigma(\gamma(\mathbf{XE} + \mathbf{W}_b^{\mathbf{E}}))\mathbf{W}^{\mathbf{L}} + \mathbf{W}_b^{\mathbf{L}}))\mathbf{D} + \mathbf{W}_b^{\mathbf{D}}). \tag{3}$$

We apply a sigmoid activation function ($\sigma$) after each layer (and in particular the output layer, wherein a softmax is *not* appropriate for multi-label data). The sigmoid function ensures bounded activations, which suits CLML's shallow and matrix-based representation. Furthermore, activation functions such as ReLU and GELU are more tailored to deeper architectures and address specific issues such as vanishing gradients, which is not as relevant in this paper. Tight-bound complexity scales linearly with the parameters in the encoding $\Theta(NDC)$ and decoding stages $\Theta(NKC)$, and quadratically with the parameters of the feedforward step $\Theta(NC^2)$. Note the complexity assumes a naive implementation of matrix multiplication. We deliberately choose a simpler and shallow model to demonstrate the effectiveness of the consistent Lebesgue measure, described in the following subsection.

## 2.3 A Lebesgue measure for surrogate-free multi-label learning

The Lebesgue measure is widely used for multi-criteria, multi-task, and multi-objective problems (Igel et al., 2007; Bader et al., 2010; Bader & Zitzler, 2011). We assume the learning task maps a batch (*i.e.*, matrix) of input vectors $\mathbf{X}^{N \times D}$ toward a batch of target labels $\mathbf{Y}^{N \times K}$. Let $\boldsymbol{\mathcal{L}}(f(\mathbf{X}), \mathbf{Y}) : \mathbb{R}^{N \times D} \to \mathbb{R}^o$ be a series of loss functions that map $\mathbf{X}$ to a vector of losses $\boldsymbol{\mathcal{L}}(f(\mathbf{X}), \mathbf{Y}) = (\mathcal{L}_1(f(\mathbf{X}), \mathbf{Y}), \cdots, \mathcal{L}_o(f(\mathbf{X}), \mathbf{Y}))$ given the objective space $Z \subseteq \mathbb{R}^o$ and a representation of a neural-network $f$. Given $o = 3$, let $\mathcal{L}_1, \mathcal{L}_2$, and $\mathcal{L}_3$ respectively, without loss of generality, represent the following widely-used multi-label loss functions: hamming loss, one minus label-ranking average precision, and one minus micro $F_1$, all of which should be minimised (see Liu et al. (2021)). Let $R \subset Z$ denote a set of mutually non-dominating[1] loss vectors; $F$, the set of representations of functions; and $H(F, R) \subseteq Z$, the set of loss vectors that dominate at least one element of $R$ and are dominated by at least one element of $F$:

$$H(F, R) := \{\mathbf{z} \in Z \quad | \quad \exists f \in F, \quad \exists \mathbf{r} \in R : \boldsymbol{\mathcal{L}}(f(\mathbf{X}), \mathbf{Y}) \prec \mathbf{z} \prec \mathbf{r}\}. \tag{4}$$

The Lebesgue measure is defined as $\lambda(H(F, R)) = \int_{\mathbb{R}^o} \mathbf{1}_{H(F,R)}(\mathbf{z})d\mathbf{z}$, where $\mathbf{1}_{H(F,R)}$ is the indicator function of $H(F, R)$. The contribution of $f$ toward the improvement (minimisation) of a set of loss functions $\boldsymbol{\mathcal{L}}$ can be quantified by first measuring the improvement of $\mathbf{x}$ via the partition function $P(f)$:

$$P(f) = H(\{f\}, R) \backslash H(F \backslash \{f\}, R). \tag{5}$$

Hence, the Lebesgue contribution of $f$, $\lambda(P(f)) = \int_{\mathbb{R}^o} \mathbf{1}_{P(f)}(\mathbf{z})d\mathbf{z}$, describes it's contribution to minimising $\boldsymbol{\mathcal{L}}$. Ultimately, $\lambda(H(F, R))$, and therefore $\boldsymbol{\mathcal{L}}$, is sought to be optimised via $\lambda(P(f))$ (see Lemma A.1), *i.e.*, $f$ is guided by evaluating its Lebesgue contribution $\lambda(P(f))$, which can be efficiently estimated using Monte Carlo sampling (Bader et al., 2010). $R$ is initialised to the unit loss vector $\{1\}^3$. Optimisation of CLML using the Lebesgue measure is achieved via covariance matrix adaptation (Hansen & Ostermeier, 1996), a standard non-convex optimiser (Smith-Miles & Geng, 2020; Sarafian et al., 2020; Nomura et al., 2021). Please refer to Section A.1 for an expanded technical exposition on covariance matrix adaptation (see Section A.1.1), Lebesgue measure estimation using Monte Carlo sampling (see Section A.1.2) and the optimisation process (see Section A.1.3 and Algorithm 1).

## 3 Consistency of the Lebesgue measure

There are many definitions of consistency including infinite-sample consistency in Zhang (2004) and edge consistency in Duchi et al. (2010). In this work, we define consistency as the Bayes risk, following Gao & Zhou (2011). We provide the following key definitions before introducing multi-label consistency.

---

[1] $f$ dominates $f'$ ($f \prec f'$) iff $\forall i : 1 \le i \le o : \mathcal{L}_i(f(\mathbf{X}), \mathbf{Y}) \le \mathcal{L}_i(f'(\mathbf{X}), \mathbf{Y})$, and $\exists \mathcal{L}_i : \mathcal{L}_i(f(\mathbf{X}), \mathbf{Y}) < \mathcal{L}_i(f'(\mathbf{X}), \mathbf{Y})$

**Definition 3.1** (Conditional Risk). *The expected conditional risk $R$, and the Bayesian risk $R^B$, of a model representation $f$ given both $\mathcal{L}$ and surrogate loss $\psi$ is defined as:*

$$R(f) = \mathbb{E}_{(\boldsymbol{x},\boldsymbol{y})\sim\mathcal{P}}[\mathcal{L}^c(p(\boldsymbol{y}|\boldsymbol{x}),f)] \quad R^B(f) = \mathbb{E}_{(\boldsymbol{x},\boldsymbol{y})\sim\mathcal{P}}[\inf_{f'}[\mathcal{L}^c(p(\boldsymbol{y}|\boldsymbol{x}),f')]]$$

$$R_\psi(f) = \mathbb{E}_{(\boldsymbol{x},\boldsymbol{y})\sim\mathcal{P}}[\psi^c(p(\boldsymbol{y}|\boldsymbol{x}),f)] \quad R_\psi^B(f) = \mathbb{E}_{(\boldsymbol{x},\boldsymbol{y})\sim\mathcal{P}}[\inf_{f'}[\psi^c(p(\boldsymbol{y}|\boldsymbol{x}),f')]]. \tag{6}$$

**Definition 3.2** (Bayes Predictors). *The set of Bayes predictors $B(p(\boldsymbol{y}|\boldsymbol{x})) = \{f : \mathcal{L}^c(p(\boldsymbol{y}|\boldsymbol{x}),f) = \inf_{f'}[\mathcal{L}^c(p(\boldsymbol{y}|\boldsymbol{x}),f')]\}$ determine that $\psi$ can be multi-label consistent w.r.t. $\mathcal{L}$ if the following holds for every $p(\boldsymbol{y}|\boldsymbol{x}) \in \kappa$:*

$$R_\psi^B(p(\boldsymbol{y}|\boldsymbol{x})) < \inf_f\{R_\psi(p(\boldsymbol{y}|\boldsymbol{x})) : \forall f \in \Omega, f \notin B\}. \tag{7}$$

**Theorem 3.1** (Multi-label Consistency). *$\psi$ can only be multi-label consistent w.r.t. $\mathcal{L}$ iff it holds for any sequence of $f^{(n)}$ that:*

$$R_\psi(f^{(n)}) \to R_\psi^B(f) \quad then \quad R_\mathcal{L}(f^{(n)}) \to R_\mathcal{L}^B(f). \tag{8}$$

The proof of Theorem 3.1 is available in Gao & Zhou (2011).

**Definition 3.3** (Pareto optimal set). *A Pareto optimal set of functions $\mathbb{P}^B$ contain the following functions:*

$$\mathbb{P}^B = \{f : \{f' : f' \prec f \quad \forall f', f \in \Omega, f' \neq f\} = \emptyset\}. \tag{9}$$

Recall that $f$ is said to dominate $f'$ ($f \prec f'$) iff $\forall i : 1 \leq i \leq o : \mathcal{L}_i(f(\mathbf{X}),\mathbf{Y}) \leq \mathcal{L}_i(f'(\mathbf{X}),\mathbf{Y})$, and $\exists \mathcal{L}_i : \mathcal{L}_i(f(\mathbf{X}),\mathbf{Y}) < \mathcal{L}_i(f'(\mathbf{X}),\mathbf{Y})$.

**Theorem 3.2** (A Consistent Lebesgue Measure). *Given a sequence $F^{(n)}$, the maximisation of the Lebesgue measure $\lambda(H(F^{(n)},R))$ is consistent with the minimisation of $\mathcal{L}_1, \mathcal{L}_2$, and $\mathcal{L}_3$:*

$$\lim_{n\to\infty} \lambda(H(F^{(n)},R)) \to \lambda(H(\mathbb{P}^B,R)) \quad then$$

$$R_{\mathcal{L}_1}(f^{(n)}) \to R_{\mathcal{L}_1}^B(f) \wedge R_{\mathcal{L}_2}(f^{'(n)}) \to R_{\mathcal{L}_2}^B(f') \wedge R_{\mathcal{L}_3}(f^{''(n)}) \to R_{\mathcal{L}_3}^B(f''). \tag{10}$$

In other words, the maximisation of $\lambda(H(F^{(n)},R))$ tends to the convergence toward the Bayes risk for each loss function $\mathcal{L}_i \; \forall i : 1 \leq i \leq 3$, $f^{(n)}, f^{'(n)}, f^{''(n)} \in F^{(n)}$ and that $f, f', f'' \in \mathbb{P}^B$. The proof of theorem 3.2 is available in Section A.6. The following section provides significant empirical evidence to support the consistency of the Lebesgue measure.

## 4 EVALUATION OF THE MULTI-LABEL CLASSIFICATION PERFORMANCE

### 4.1 COMPARATIVE STUDIES

We compare CLML against several state-of-the-art and benchmark methods using the recommended parameter configurations in their papers.

Our first comparative method is, to the best of our knowledge, the current state-of-the-art and best tabulated multi-label learner to date: collaborative learning of label semantics and deep label-specific features (CLIF) (Hang & Zhang, 2022). CLIF exploits label interactions using label graph embedding. Label graphs are encoded using a graph isomorphism network, where the respective embeddings are used to weigh the latent representation of each instance. Label graph embeddings are adjusted during the learning process via backpropagation. CLIF has achieved state-of-the-art results in comparison to many well-known tabulated multi-label learners (Zhang & Wu, 2014; Huang et al., 2015; 2017; Yeh et al., 2017; Ma & Chow, 2020; Bai et al., 2021).

Our second comparative method learns deep latent spaces for multi-label classification (C2AE) (Yeh et al., 2017), which jointly encodes features and labels into a shared semantic space via an autoencoder network. Our third comparison method is ML$k$NN, which is a multi-label variant of $k$NN that estimates labels based on Bayesian inference (Zhang & Zhou, 2007). The final two comparison methods are based on Gaussian Naive Bayes with binary relevance (GNB-BR) and classifier chain (GNB-CC) transformations (Read et al., 2011).

Table 1: Averages and medians of the geometric means, for all methods across all datasets.

| $f$ | $\mu(\mu_g(\boldsymbol{\mathcal{L}}(f)))$ | $Med(\mu_g(\boldsymbol{\mathcal{L}}(f)))$ |
|---|---|---|
| GNB-BR | $0.381 \pm 0.130$ | 0.371 |
| GNB-CC | $0.361 \pm 0.096$ | 0.376 |
| ML$k$NN | $0.246 \pm 0.087$ | 0.239 |
| C2AE | $0.466 \pm 0.233$ | 0.419 |
| CLIF | $\mathbf{0.232} \pm 0.087$ | 0.269 |
| CLML | $\mathbf{0.232} \pm 0.073$ | **0.237** |

Table 2: Friedman statistic of each measure for all comparative methods across all datasets.

| Measure | Statistic | Critical Value |
|---|---|---|
| $\lambda(P(f))$ | 2.542 | |
| $\mu_g(\boldsymbol{\mathcal{L}}(f))$ | 30.000 | |
| $\mathcal{L}_1(f)$ | 39.022 | 15.507 |
| $\mathcal{L}_2(f)$ | 26.222 | |
| $\mathcal{L}_3(f)$ | 29.600 | |

## 4.2 STATISTICAL ANALYSIS

We compare CLML against CLIF, C2AE, ML$k$NN, GNB-CC, and GNB-BR using several multi-label loss functions on nine different, commonly used open-access datasets. The experimental protocol and dataset summary are described in detail in Section A.2.

### 4.2.1 AGGREGATED PERFORMANCES

The geometric mean of the loss vector $\boldsymbol{\mathcal{L}}(f(\mathbf{X}), \mathbf{Y})$, $\mu_g(\boldsymbol{\mathcal{L}}(f(\mathbf{X}), \mathbf{Y})) = (\prod_{i=1}^{3} \mathcal{L}_i(f(\mathbf{X}), \mathbf{Y}))^{\frac{1}{3}}$, determines the aggregate performance of the loss functions. Owing to its multiplicative nature, the geometric mean can be sensitive to very low values among the loss functions, which can result in a lower aggregate value than the arithmetic mean, which accords each value with equal importance. This sensitivity grants the geometric mean with finer granularity, which is beneficial when: (1) the range and magnitude of values among the loss functions are comparable in scale, and (2) distinguishing solutions that exhibit exemplary performance on a subset of the loss functions is a priority. Table 1 includes the summary values of the geometric means over all datasets for each of the comparative methods.

CLML achieves the lowest (median) geometric mean, indicating the best performance when balancing all three loss functions. Even though CLML and CLIF achieve the same average score, the geometric mean variability of CLML is lower by $16.1\%$, indicating that CLML achieves the best performance more consistently.

### 4.2.2 NON-PARAMETRIC TESTS

We employ the widely-used non-parametric Friedman test Demšar (2006) to measure any statistically significant performance differences between the methods with respect to the Lebesgue contribution, the geometric mean, and each individual loss function. Table 2 exhibits the significant differences between the geometric means and each individual loss function across all comparative methods and datasets. Interestingly, Table 2 shows that the Friedman test of the Lebesgue contributions did not reflect these differences. Note that these observations do not indicate uniformity in the performance of all methods. For example, while different methods may contribute solutions of different qualities, said solutions may intersect in terms of their Lebesgue contributions. This is evident in the supplementary Tables 6 and 7, where all dominated methods have a Lebesgue contribution of zero. For this reason, we propose another analysis.

We proceed to further analyse model performance using a Bonferroni-Dunn test and a critical difference analysis with $\alpha = 0.05$ (Dunn, 1961; Hang & Zhang, 2022). We perform a pairwise comparison between the average ranks of each algorithm, with CLML set as a control algorithm. The critical difference plots in Figure 2 illustrate the rankings of each algorithm. The rankings are presented in ascending order, where the best method is on the leftmost side of the plot.

In summary, CLML achieves the lowest aggregate rank of $1.94$ (aggregated among all measures and datasets presented in Figure 2), compared to $2.24$ of CLIF ($+13.4\%$), $3.1$ of ML$k$NN ($+37.4\%$), $4.36$ of C2AE ($+55.5\%$), $4.6$ of GNB-CC ($+57.8\%$), and $4.78$ of GNB-BR ($+59.4\%$). The only ranking that CLML is not the best at is with respect to $\mathcal{L}_1$ (losing to CLIF by $0.2$). CLML and CLIF obtain the same ranking with respect to $\mathcal{L}_3$. Furthermore, in most cases, the Bonferroni-Dunn test does not detect a significant difference between CLML and CLIF, even though CLML achieves a better rank than CLIF in most cases. This largely stems from CLIF being the current state-of-the-art, remaining competitive with CLML in most cases.

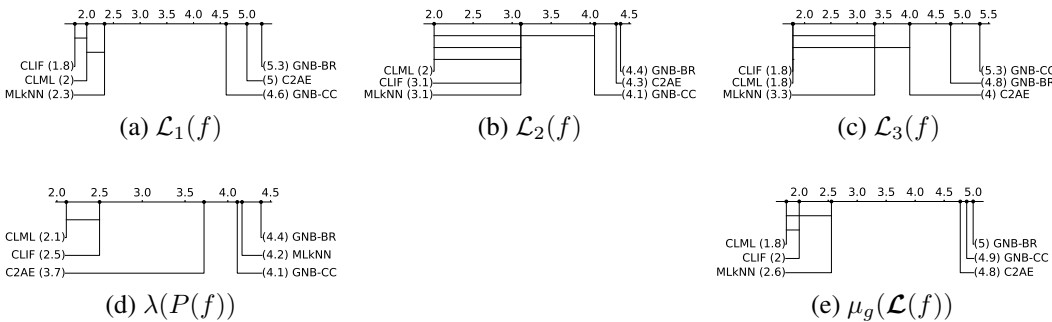

Figure 2: Bonferroni-Dunn test critical difference plots. A crossbar is drawn between CLML and any method if their difference in average ranking is less than the critical difference ($CD = 2.266$ with $K = 6$ methods and $T = 9$ datasets).

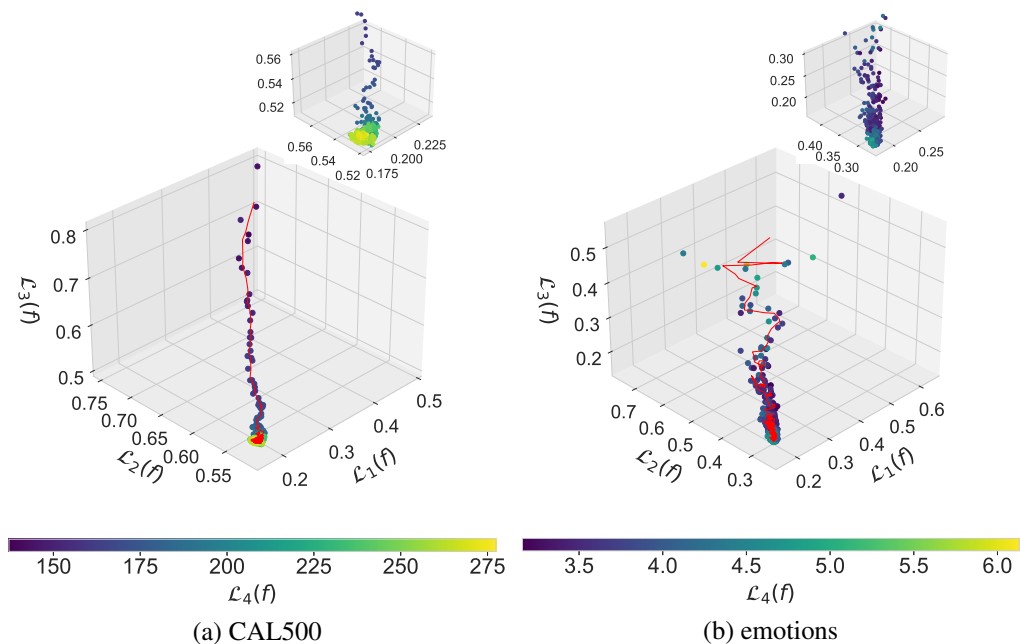

Figure 3: The training curves of CLML plotted against $\mathcal{L}_1(f(\mathbf{X}), \mathbf{Y})$, $\mathcal{L}_2(f(\mathbf{X}), \mathbf{Y})$, and $\mathcal{L}_3(f(\mathbf{X}), \mathbf{Y})$. The colour represents the averaged binary cross-entropy loss $\mathcal{L}_4(f(\mathbf{X}), \mathbf{Y})$, which is tracked independently during the optimisation process. The red line shows the moving average trajectory of CLML. A zoom-in plot is presented at the top right of each subplot to highlight the area of convergence.

These results are significant due to the simple feedforward representation of CLML, which does not include the label graph embedding or semantic embedding techniques seen in CLIF or C2AE, respectively. This highlights the influence and importance of a consistent loss function that properly directs optimisation behaviour and quality of multi-label learners. To further support this claim, the next section provides further analysis to examine the relationship between the average binary cross entropy and the loss functions that are optimised by CLML.

## 5 EFFECTIVENESS OF THE LEBESGUE MEASURE OVER SURROGATE LOSS

This section analyses the relationship between the loss functions: $\mathcal{L}_1$, $\mathcal{L}_2$, and $\mathcal{L}_3$, and the averaged binary cross entropy loss representing a standard surrogate loss function (Yeh et al., 2017; Bai et al., 2021; Hang & Zhang, 2022), denoted $\mathcal{L}_4$. The following key observations can be made.

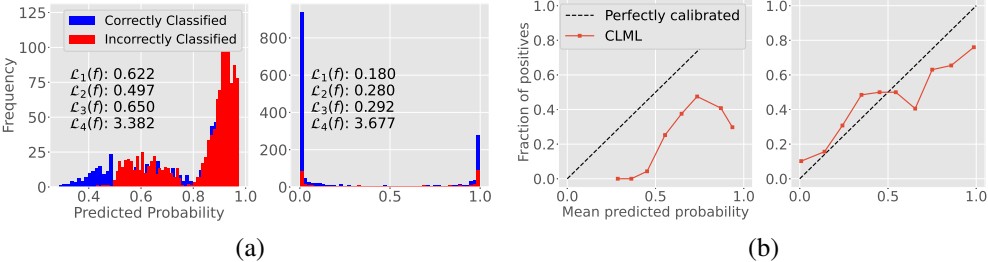

Figure 4: Label distributions and calibration curves of CLML on the emotions dataset. Subplot (a) shows two label probability distributions. The leftmost plot is the distribution of CLML's incumbent label probabilities after 1 epoch. The plot to the right is the distribution of the predicted label probabilities after training is complete. Subplot (b) shows the calibration curve of the incumbent solution of CLML after the first epoch (left) and its state after training is complete (right). The number of bins for the distributions and the calibration plots are $B_D = 50$ and $B_C = 10$, respectively.

**Moving average trajectory.** Figure 3 plots the training trajectory of CLML with respect to the four loss functions during training on the CAL500 and emotions datasets. The remaining training curves are available in Figures 7 and 8 in Section A.5. The red line traces the moving average trajectory of CLML on the approximate loss landscape defined by the three loss functions $\mathcal{L}_1$, $\mathcal{L}_2$, and $\mathcal{L}_3$, while the point colour denotes the $\mathcal{L}_4$ loss.

The moving average trajectory indicates a distinct and consistent decline of all the desired loss functions $\mathcal{L}_1$, $\mathcal{L}_2$, and $\mathcal{L}_3$. Recall that CLML specifically optimises the Lebesgue *contribution* $\lambda(P(f))$, *i.e.*, the contribution toward the improvement of the Lebesgue measure $\lambda(H(F, R))$. In this case, by empirical observation, maximising the Lebesgue contribution directly corresponds to the minimisation of the desired loss functions. Furthermore, despite a degree of stochasticity, the overarching trend indicates that all three loss functions can be optimised in tandem despite their conflicting nature (see Wu & Zhu (2020) for an analysis of the conflicting nature of multi-label loss functions). This is primarily due to the inherent nature of the Lebesgue measure, which naturally considers the trade-offs between the different multi-label loss functions. Hence, the optimisation behaviour of CLML naturally follows a path that understands and accounts for the best trade-offs between these loss functions. Furthermore, Figure 3 draws attention to the increase in $\mathcal{L}_4$ near the point of convergence on both CAL500 and emotions. On both datasets, the minimisation of $\mathcal{L}_1$, $\mathcal{L}_2$, and $\mathcal{L}_3$ does not directly correspond to the minimisation of $\mathcal{L}_4$, which suggests a discrepancy between the surrogate and desired loss functions.

**Label confidence vs. label accuracy.** Figures 4(a) and (b) plot the distributions of CLML's incumbent label probabilities and the calibration curve of CLML's incumbent solution respectively, after one epoch (left) and after training (right).

We first discuss the distributions after one epoch. Both correct and incorrect label probability distributions are uni-modal and share a similar shape, with noticeable variation in confidence ranging from 0.5 to 0.8. This suggests that CLML is initially less confident in its prediction of the absence of labels than its prediction of the presence of labels (irrespective of correctness). After training, the distributions are heavily skewed and bi-modal, which suggests that CLML is very confident in predicting labels. This bi-modal shape applies to both the correct and incorrect label distributions. Notably, the value of $\mathcal{L}_4$ after one epoch increases from 3.382 to 3.677 after training, an undesired effect. This increase can be attributed to CLML's increased degree of confidence in the incorrectly classified labels. However, $\mathcal{L}_1$, $\mathcal{L}_2$, and $\mathcal{L}_3$ are desirably lower after training, which is further supported by the improvement in the calibration curve of CLML after training in Figure 4(b). These observations highlight the discrepancy between confidence and accuracy, which underscores the importance of directly handling accuracy (without surrogacy) in multi-label learning.

## 6 RELATED WORKS

**Multi-label Learning**. Multi-label learning is a common problem, including computer vision (You et al., 2020; Wang et al., 2020; Zhou et al., 2021; Yuan et al., 2023; Liu et al., 2023), functional ge-

nomics (Patel et al., 2022), and tabulated learning (Yeh et al., 2017; Bai et al., 2021; Hang & Zhang, 2022; Lu et al., 2023). Earlier work on multi-label learning transformed a multi-label problem into a series of single-label problems, or by sequentially classifying each label, where previously classified labels are incorporated at each classification step (Liu et al., 2021). However, such a transformation does not typically perform well, as valuable label interactions are lost. CLML's universal performance gain over both GNB-BR and GNB-CC highlights this property of multi-label learning, as it does not transform the multi-label problem into a single-label problem.

Deep learning has advanced the multi-label learning field (Liu et al., 2017). Researchers have followed this trend, exploiting feature interactions using self-attention mechanisms on transformers (Xiao et al., 2019), and deep latent space encoding of features and labels using auto-encoders (C2AE) (Yeh et al., 2017). CLML demonstrates that a relatively *straightforward* architecture, without such deep encoding mechanisms, can be state-of-the-art. This highlights the overarching importance of the consistent Lebesgue measure to multi-label learning.

Collaborative learning of label semantics and deep label-specific features (CLIF) (Hang & Zhang, 2022) exploits label interactions using label graph embedding. CLIF has achieved state-of-the-art results in multi-label learning, significantly outperforming many prior state-of-the-art methods. CLIF utilises a label graph embedding procedure that weighs the latent representation of the input. With a simple representation, CLML achieves an overall lower ranking (better) than CLIF across a variety of loss functions and datasets (a $13.4\%$ improvement). This is particularily significant given the state-of-the-art performance of CLIF, which significantly outperforms many prior state-of-the-art works such as LIFT (Zhang & Wu, 2014), LLSF (Huang et al., 2015), JFSC (Huang et al., 2017), C2AE (Yeh et al., 2017), TIFS (Ma & Chow, 2020), and MPVAE (Bai et al., 2021).

**Consistency**. All multi-label loss functions are ultimately related to measuring the predicted label accuracy. However, their interpretation of quality, and therefore their consistency, can vary. A robust multi-label model should account for the inconsistencies between the multiple related, yet potentially conflicting, loss functions in multi-label learning (Gao & Zhou, 2011; Wu & Zhu, 2020; Liu et al., 2021). Consistency in multi-label learning has been investigated in prior works Gao & Zhou (2011) and Wu & Zhu (2020), showing both that we can only partially approximate some loss functions and that feasibility of a consistent loss approximation remains an open-ended research question. In this paper, we have theoretically and empirically shown that the Lebesgue measure, over a series of multi-label loss functions ($\mathcal{L}_1$, $\mathcal{L}_2$, and $\mathcal{L}_3$), is consistent with the loss functions themselves. As a result, CLML circumvents the issues surrounding surrogate loss functions.

# 7 CONCLUSIONS

Deep learning techniques have advanced the field of multi-label learning, and have thus become state-of-the-art. Despite the success achieved by deep learning-based multi-label learning methods, inconsistencies remain between surrogate loss functions and multi-label functions and have seldom been addressed. This motivates the proposed Consistent Lebesgue Measure-based Multi-label Learner (CLML). In this paper, we proposed a highly novel approach to tabulated multi-label learning that considers multiple loss functions simultaneously. These multiple non-convex and discontinuous loss functions are optimised using a novel Lebesgue measure-based learning objective. By analysis, we proved that CLML is theoretically Bayes consistent with the underlying loss functions that are optimised. Furthermore, empirical evidence supports our theory by demonstrating a $13.4\%$ to $59.4\%$ improvement in the critical distance rankings of CLML in comparison to state-of-the-art methods. These results are especially significant due to the *simplicity* of CLML, which achieves state-of-the-art results without the need to explicitly consider label interactions via label graphs or latent semantic embeddings. Lastly, our analysis shows that CLML can naturally account for the best trade-offs between multiple multi-label loss functions that are known to exhibit conflicting behaviour. CLML's state-of-the-art performance further highlights the importance of optimisation consistency over model complexity. Thus, the findings of this paper analytically emphasises the overall significance of consistency and goal alignment in multi-label learning.

## 8 REPRODUCIBILITY STATEMENT

The code and pre-trained weights for CLML will be made available upon submission. All datasets are open access at `https://mulan.sourceforge.net/datasets-mlc.html`.

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

## A  APPENDIX

### A.1  OPTIMISATION PROCESS

The overall optimisation procedure using covariance matrix adaptation and the Lebesgue measure is described clearly in the following sections.

#### A.1.1  COVARIANCE MATRIX ADAPTATION

Co-variance Matrix Adaptation Evolutionary Strategy (CMA-ES) Hansen & Ostermeier (2001) is a gradient-free numerical optimisation technique well-suited for non-convex and non-differentiable optimisation problems. Suppose the representation of a learner $f$ can be denoted as a vector consisting of its learnable parameters $\theta^f$. CMA-ES works by sampling $\lambda$ solutions from a multi-variate normal distribution as follows:

$$\theta_i^f \sim \mathbf{m} + \sigma \mathcal{N}_i(0, \mathbf{C}) \quad \forall i, \quad 1 \leq i \leq \lambda \tag{11}$$

where $\theta_i^f$ is the learnable parameters of the $i^{th}$ learner, $\mathbf{m}$ is a mean-vector which represents the expected density of parameters of $f$, $\sigma$ the step-size, and $\mathbf{C}$ the covariance matrix. CMA-ES therefore iteratively updates $\mathbf{m}$ and $\mathbf{C}$ via the following:

$$\mathbf{m}^{t+1} = \mathbf{m}^t + \sigma \sum_{i=1}^{\mu} w_i \theta_i^{f^{top}} \tag{12}$$

$$\mathbf{C}^{t+1} = (1 - c_{cov})\mathbf{C}^t + c_{cov} \sum_{i=1}^{\mu} w_i \theta_i^{f^{top}} (\sum_{i=1}^{\mu} w_i \theta_i^{f^{top}})^T \tag{13}$$

$$\tag{14}$$

where $c_{cov}$ is the learning rate, $\sum_{i=1}^{\mu} w_i \theta_i^{f^{top}}$ is the weighted sum of the $\mu$-highest ranked solutions at time $t$, where the weights $w_1 > w_2 > \cdots > w_\mu > 0$ and $\sum_{i=1}^{\mu} w_i = 1$. It is also deemed that solutions $\theta_i^{f^{top}} \sim \mathbf{m} + \sigma \mathcal{N}_i(0, \mathbf{C})$ are ranked such that $\theta_1^{f^{top}} \prec \cdots \prec \theta_\mu^{f^{top}}$ and that the $\mu$ ranked solutions are a subset of the total number of sampled solutions, i.e. $\mu < \lambda$. This method is referred to rank-one update. CMA-ES can also be updated using the cumulation (or search trajectory), hence the covariance matrix is updated as follows using the rank-one method:

$$\mathbf{C}^{t+1} = (1 - c_{cov})\mathbf{C}^t + c_{cov} \sum_{i=1}^{\mu} w_i \mathbf{p}_c^t \mathbf{p}_c^{t\,T} \tag{15}$$

$$\mathbf{p}_c^{t+1} = (1 - c_c)\mathbf{p}_c^t + \sqrt{1 - (1 - c_c)^2} \sqrt{\mu_{eff}^t} \sum_{i=1}^{\mu} w_i \theta^{f\,top} \tag{16}$$

$$\mu_{eff}^t = \frac{1}{\sqrt{\sum_{i=1}^{\mu} w_i^2}} \sum_{i=1}^{\mu} w_i \theta^{f\,top} \tag{17}$$

where $c_c$ is the prescribed learning rate for the cumulation update.

#### A.1.2  LEBESGUE MEASURE ESTIMATION USING MONTE CARLO SAMPLING

The Lebesgue measure $\lambda(H(F, R))$ described in Eq. 4 integrates the area covered by a set of loss function vectors in a multi-dimensional objective space. This measure is comprised of three sets: $F$, $R$, and $Z$. $F$ denotes the set of representations of functions (which map the input data to a vector of loss function values). $R$ denotes the set of mutually non-dominating loss vectors. Initially, $R$ is set to the unit loss vector $\{1\}^3$, which denotes the worst possible performance for Hamming-loss, one minus the Micro-$F_1$, and one minus the label ranking average precision. Last, $Z$ denotes the set containing all possible loss function vectors in the applicable multi-dimensional loss objective space.

The Lebesgue contribution $\lambda(P(f))$ of a function $f$ measures the new marginal improvement of a function's loss vector over a set of previous loss vectors. In this paper, we use the Lebesgue contribution to quantify candidate functions found by CLML during the optimisation process. However,

to efficiently calculate the Lebesgue contribution (especially when the set of functions $F$ and $R$ are sparsely populated during the early stages of the optimisation), we estimate the Lebesgue measure using Monte Carlo sampling. First, a sampling space $S \subseteq Z$ is defined that entirely contains $P(f)$, *i.e.,* $P(f) \subseteq S \subseteq Z$. The sampling space can be problem-specific, however, in this paper, it is defined to contain all possible loss vectors between $\{0\}^3$ and $\{1\}^3$. Following, $g$ samples are drawn from $s_i \in S$ randomly and with uniform probability. Given $\{s_1, \cdots, s_g\}$, the Lebesgue contribution is estimated via $\hat{\lambda}(P(f))$ via the following:

$$\hat{\lambda}(P(f)) = \lambda(S(f)) = \frac{|\{s_i | s_i \in P(f)\}|}{g} \tag{18}$$

where $|\{s_i | s_i \in P(f)\}|$ is denoted as the number of randomly sampled solutions that exist in $P(f)$, also known as *hits*. The probability $p$ of a sample being *hit* is i.i.d. Bernoulli distributed, therefore, $\hat{\lambda}(P(f))$ converges to $\lambda(P(f))$ with $\frac{1}{\sqrt{pg}}$ Laplace (1814).

### A.1.3 Optimisation of the Lebesgue measure using covariance matrix adaptation

The optimisation process is described in Algorithm 1. Starting with an initial covariance matrix and density vector, CLML optimises the Lebesgue contribution of newly generated candidate functions obtained by perturbing a density vector and covariance matrix. Each function is evaluated using the Hamming-loss ($\mathcal{L}_1$), one minus the label ranking average precision ($\mathcal{L}_2$), one minus the micro-$F_1$ ($\mathcal{L}_3$). The density vector is updated toward the density of the current solutions, and the covariance matrix is updated using a rank-one method. CLML maintains an archive of solutions based on the validation fitness values that are derived from the prescribed loss functions. Ultimately, CLML returns the incumbent solutions (in terms of validation loss) for each of the loss functions from the archive and the final incumbent solution.

---

**Algorithm 1:** Consistent Lebesgue Measure-based Multi-label Learner

---

**Input:** Initial covariance matrix $\mathbf{C}^0$, and density vector $\mathbf{m}^0$, and maximum number of epoch $T$

Initialise $R^0$ to unit vector $\{1\}^3$;
Initialise $F^0 = \{\emptyset\}$;
Set $t = 0$;
Set $f^0 = \mathbf{m}^0$;
**while** $t < T$ **do**

    Generate $f^i \sim \mathbf{m}^t + \sigma \mathcal{N}_i(0, \mathbf{C}^t)$, $1 \leq i \leq \lambda$;

    Set $F^{t+1} = \bigcup_{i=1}^{\lambda} \{f^i\}$;

    **for** $f^i \in F^{t+1}$ **do**

        Calculate the training ($tra$) and validation ($val$) loss values for each loss function: $\mathcal{L}_1(f^i)$, $\mathcal{L}_2(f^i)$, and $\mathcal{L}_3(f^i)$;

        Estimate $\lambda(P(f^i))$ using the Monte Carlo method over loss functions $\mathcal{L}_1^{tra}(f^i)$, $\mathcal{L}_2^{tra}(f^i)$, and $\mathcal{L}_3^{tra}(f^i)$, and prescribe it as the fitness for $f^i$;

        Archive the loss values $\mathcal{L}_1^{val}(f^i)$, $\mathcal{L}_2^{val}(f^i)$, and $\mathcal{L}_3^{val}(f^i)$, and corresponding function $f^i$;

    Update density $\mathbf{m}^{t+1}$ to the average of the newly generated solutions $\forall f^i \in F^{t+1}$;

    Update $\mathbf{C}^{t+1}$ via rank-one method using the prescribed $\lambda(P(f^i))$ as fitness values $\forall f^i \in F^{t+1}$;

    Update $R^{t+1}$ by calculating the mutually non-dominated solutions in $R^t \cup F^{t+1}$;

    Set $f^{t+1}$ to the best solution in $F^{t+1}$ according to its prescribed fitness value;

**return** *Incumbent solutions for each loss function from archive:* $\mathcal{L}_1^{val}(f^i)$, $\mathcal{L}_2^{val}(f^j)$, *and* $\mathcal{L}_3^{val}(f^k)$, *and the final incumbent solution* $f^T$;

---

## A.2 EXPERIMENTAL PROTOCOL

We conduct the experiments on nine widely-used multi-label datasets, shown in Table 3. $K^\mu$ (the cardinality) of an instance measures the average number of associated class labels; $DK/K^\mu$, the theoretical maximum complexity of an instance, (*i.e.*, the instance-level average dispersion of feature to label interactions); and $DK^\mu$, the average feature to label interactions of an instance. There are two important cases to consider. First, if dispersion is less than the average interaction, *i.e.*, $DK/K^\mu < DK^\mu$, then the dataset contains high concentrations of rich instance-level feature-to-label interactions that are not apparent when examining the dataset as a whole. This can indicate that there are clusters of instances that share similar feature-to-label interactions, and therefore a less diverse dispersion of the possible feature-to-label interactions. Second, if dispersion is higher than the average interaction, *i.e.*, $DK/K^\mu > DK^\mu$, the dataset as a whole has a greater expression of feature-to-label interactions than a given individual instance. Put differently, the dataset's individual instances each contain a subset of the total dataset interactions. The latter case is particularly challenging as it indicates a high number and variability of potential patterns and interactions between features and labels. The first case occurs in both Flags and Yeast and the second case occurs in the remaining datasets.

For each dataset, 30% are partitioned to the test set (Sechidis et al., 2011). The remaining 70% is further split such that 20% is used as a validation set, and the remaining is used for training. We apply normalisation to all numerical features before training.

Table 3: Summary of datasets used in this paper. $D$, $N$, and $K$ correspond to the number of features, instances, and labels, respectively.

| Dataset | $D$ | $N$ | $K$ | $DK$ | $K^\mu$ | $DK/K^\mu$ | $DK^\mu$ |
|---|---|---|---|---|---|---|---|
| flags | 19 | 194 | 7 | 133 | 3.392 | 39.21 | 64.45 |
| CAL500 | 68 | 502 | 174 | 251,000 | 26.044 | 9,637.54 | 1,770.99 |
| emotions | 72 | 593 | 6 | 432 | 1.869 | 231.14 | 134.57 |
| yeast | 103 | 2417 | 14 | 1442 | 4.237 | 340.335 | 436.411 |
| scene | 294 | 2407 | 6 | 1764 | 1.074 | 1,642.46 | 315.756 |
| corel5k | 499 | 5000 | 374 | 186,626 | 3.522 | 52,988.64 | 1,757.48 |
| enron | 1001 | 1702 | 53 | 53,053 | 3.378 | 15,705.45 | 3,381.38 |
| genbase | 1186 | 662 | 27 | 32,022 | 1.252 | 25,576.68 | 1,484.87 |
| medical | 1449 | 978 | 45 | 65,205 | 1.245 | 52,373.49 | 1,804.01 |

## A.3 ABLATION STUDY

We trial the embedding dimension $C$ at eight separate values. It is important to note that the latent space does not need to express spatial relationships of tabulated data, hence the embedding dimension can be quite small (in contrast to computer vision in works such as Gong et al. (2022)). In addition to $\mathcal{L}_1, \mathcal{L}_2$, and $\mathcal{L}_3$, we set $\mathcal{L}_4$ as the averaged binary cross-entropy loss and track its progress during optimisation. For each experiment, we set $\mathcal{O} = 750$ (the maximum number of epochs). Here, we present the results for each of the embedding dimensions.

Figures 5 and 6 plot the Lebesgue measure of the sequence of functions obtained by CLML as $n \to \mathcal{O}$ (*i.e.*, the archive of non-dominated solutions obtained by CLML in $\mathcal{O}$ epoch). Smaller embedding dimensions (*i.e.*, $C \le 80$) result in the best validation scores of $\lambda(H(F, R))$. To exemplify this, we tabulate the incumbent solution of the function sequence in terms of its $\mathcal{L}_1, \mathcal{L}_2$, and $\mathcal{L}_3$ scores on the validation set, against $\mathcal{L}_4$ according to each (non-normalised) value of $C$ in Table 4 and 5. When $C = 20$, we observe the lowest $\mathcal{L}_1, \mathcal{L}_2$, and $\mathcal{L}_3$ validation loss scores on the emotions dataset, and the lowest $\lambda(H(F, R))$ score on the CAL500 dataset. This observation indicates that CLML converges toward a better approximation of the Bayes predictors of $\mathcal{L}_1, \mathcal{L}_2$, and $\mathcal{L}_3$ on the emotions dataset, while on CAL500, CLML finds functions with more desirable trade-offs between the variant loss functions, hence the higher Lebesgue measure. These values motivate our recommendation to set the number of embedding dimensions to $C = 20$.

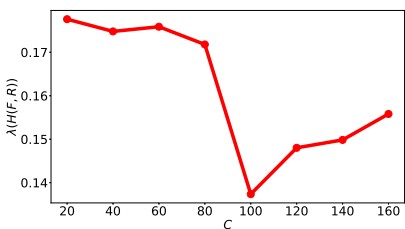

Figure 5: The best Lebesgue measure obtained on CAL500 at each embedding dimension of the sequence of function sets $\lim_{n \to \mathcal{O}} \lambda(H(F^{(n)}, R))$.

Table 4: Best validation loss values of the incumbent solution for each embedding dimension on CAL500.

| $C$ | $\mathcal{L}_1$ | $\mathcal{L}_2$ | $\mathcal{L}_3$ | $\mathcal{L}_4$ |
|---|---|---|---|---|
| 20.0 | 0.169 | 0.523 | **0.509** | **138.068** |
| 40.0 | 0.171 | **0.522** | 0.518 | 143.809 |
| 60.0 | 0.169 | 0.523 | 0.520 | 144.201 |
| 80.0 | **0.161** | 0.527 | 0.525 | 149.604 |
| 100.0 | 0.196 | 0.529 | 0.534 | 157.877 |
| 120.0 | 0.171 | 0.529 | 0.539 | 155.555 |
| 140.0 | 0.167 | 0.528 | 0.534 | 151.250 |
| 160.0 | 0.168 | 0.526 | 0.533 | 153.533 |

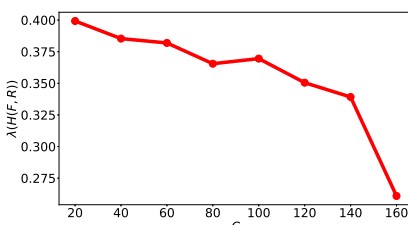

Figure 6: The best Lebesgue measure obtained on emotions at each embedding dimension of the sequence of function sets $\lim_{n \to \mathcal{O}} \lambda(H(F^{(n)}, R))$.

Table 5: Best validation loss values of the incumbent solution for each embedding dimension on emotions.

| $C$ | $\mathcal{L}_1$ | $\mathcal{L}_2$ | $\mathcal{L}_3$ | $\mathcal{L}_4$ |
|---|---|---|---|---|
| 20.0 | **0.187** | **0.283** | **0.178** | 3.399 |
| 40.0 | 0.199 | 0.307 | 0.197 | **3.246** |
| 60.0 | 0.192 | 0.299 | 0.196 | 3.255 |
| 80.0 | 0.196 | 0.306 | 0.196 | 3.910 |
| 100.0 | 0.199 | 0.302 | 0.199 | 3.742 |
| 120.0 | 0.210 | 0.333 | 0.212 | 3.557 |
| 140.0 | 0.202 | 0.313 | 0.193 | 4.380 |
| 160.0 | 0.250 | 0.398 | 0.252 | 5.758 |

### A.4 EXTENDED EVALUATION OF MULTI-LABEL CLASSIFICATION PERFORMANCES

Tables 6 and 7 show the expanded view of the loss values ($\mathcal{L}_1$, $\mathcal{L}_2$, and $\mathcal{L}_3$), the Lebesgue contribution ($\lambda(P(f))$), the normalised Lebesgue contribution, and geometric means of each comparative method on each dataset. A zero value on the Lebesgue contribution indicates that a given function is dominated by all other functions on the given dataset, *i.e.*, it does not contribute toward the improvement of the volume over $\mathcal{L}(f(\mathbf{X}), \mathbf{Y})$.

Table 6: Lebesgue measure contribution of each $f$ on datasets: CAL500 to enron

| Dataset | $f$ | $(\mathcal{L}_1, \mathcal{L}_2, \mathcal{L}_3)$ | $\lambda(P(f))$ | Normalised $\lambda(P(f))$ | $\mu_g(\mathcal{L}(f(\mathbf{X}), \mathbf{Y}))$ |
|---|---|---|---|---|---|
| CAL500 | GNB-BR | (0.547, 0.713, 0.647) | 0 | 0 | 0.631976 |
| CAL500 | GNB-CC | (0.273, 0.641, 0.749) | 0 | 0 | 0.507895 |
| CAL500 | ML$k$NN | (0.150, 0.637, 0.554) | 0.000351 | 0.020338 | 0.375585 |
| CAL500 | C2AE | (0.258, 0.536, 0.534) | 0 | 0 | 0.419409 |
| CAL500 | CLIF | (0.137, 0.681, 0.502) | 0.007068 | 0.409162 | 0.360752 |
| CAL500 | CLML | (0.168, 0.526, 0.520) | 0.009856 | 0.570499 | 0.358231 |
| yeast | GNB-BR | (0.319, 0.472, 0.351) | 0 | 0 | 0.375014 |
| yeast | GNB-CC | (0.325, 0.488, 0.428) | 0 | 0 | 0.407721 |
| yeast | ML$k$NN | (0.213, 0.375, 0.298) | 0 | 0 | 0.287821 |
| yeast | C2AE | (0.221, 0.358, 0.272) | 0.003081 | 0.425447 | 0.278355 |
| yeast | CLIF | (0.227, 0.391, 0.275) | 0 | 0 | 0.290108 |
| yeast | CLML | (0.211, 0.364, 0.266) | 0.004160 | 0.574553 | 0.273480 |
| corel5k | GNB-BR | (0.035, 0.808, 0.899) | 0 | 0 | 0.293086 |
| corel5k | GNB-CC | (0.023, 0.796, 0.900) | 0.000374 | 0.050347 | 0.256293 |
| corel5k | ML$k$NN | (0.012, 0.878, 0.808) | 0 | 0 | 0.202247 |
| corel5k | C2AE | (0.027, 0.800, 0.730) | 0.001417 | 0.190763 | 0.251250 |
| corel5k | CLIF | (0.010, 0.820, 0.701) | 0.005621 | 0.756548 | 0.179956 |
| corel5k | CLML | (0.021, 0.801, 0.798) | 0.000017 | 0.002343 | 0.236797 |
| enron | GNB-BR | (0.198, 0.725, 0.776) | 0 | 0 | 0.481206 |
| enron | GNB-CC | (0.125, 0.639, 0.742) | 0 | 0 | 0.390116 |
| enron | ML$k$NN | (0.056, 0.529, 0.436) | 0 | 0 | 0.238964 |
| enron | C2AE | (0.189, 0.665, 0.487) | 0 | 0 | 0.393941 |
| enron | CLIF | (0.053, 0.499, 0.381) | 0.014309 | 0.702718 | 0.216576 |
| enron | CLML | (0.054, 0.488, 0.411) | 0.006053 | 0.297282 | 0.220966 |

Table 7: Lebesgue measure contributions of each $f$ on datasets: genbase to medical

| Dataset | $f$ | $(\mathcal{L}_1, \mathcal{L}_2, \mathcal{L}_3)$ | $\lambda(P(f))$ | Normalised $\lambda(P(f))$ | $\mu_g(\mathcal{L}(f(\mathbf{X}), \mathbf{Y}))$ |
|---|---|---|---|---|---|
| genbase | GNB-BR | (0.052, 0.479, 0.538) | 0 | 0 | 0.237314 |
| genbase | GNB-CC | (0.058, 0.639, 0.575) | 0 | 0 | 0.277332 |
| genbase | ML$k$NN | (0.033, 0.454, 0.331) | 0 | 0 | 0.170749 |
| genbase | C2AE | (0.345, 0.823, 0.561) | 0 | 0 | 0.542500 |
| genbase | CLIF | (0.046, 0.793, 0.539) | 0 | 0 | 0.269161 |
| genbase | CLML | (0.020, 0.239, 0.117) | 0.305516 | 1.000000 | 0.082065 |
| scene | GNB-BR | (0.233, 0.431, 0.351) | 0 | 0 | 0.327976 |
| scene | GNB-CC | (0.208, 0.485, 0.360) | 0 | 0 | 0.331290 |
| scene | ML$k$NN | (0.096, 0.277, 0.193) | 0 | 0 | 0.172696 |
| scene | C2AE | (0.153, 0.439, 0.240) | 0 | 0 | 0.252437 |
| scene | CLIF | (0.085, 0.244, 0.125) | 0.077641 | 1.000000 | 0.137482 |
| scene | CLML | (0.124, 0.358, 0.192) | 0 | 0 | 0.204266 |
| emotions | GNB-BR | (0.410, 0.458, 0.271) | 0 | 0 | 0.370604 |
| emotions | GNB-CC | (0.363, 0.469, 0.313) | 0 | 0 | 0.376465 |
| emotions | ML$k$NN | (0.268, 0.497, 0.361) | 0 | 0 | 0.363696 |
| emotions | C2AE | (0.537, 0.556, 0.488) | 0 | 0 | 0.526199 |
| emotions | CLIF | (0.223, 0.412, 0.246) | 0 | 0 | 0.282547 |
| emotions | CLML | (0.205, 0.328, 0.224) | 0.070462 | 1.000000 | 0.246669 |
| flags | GNB-BR | (0.443, 0.560, 0.439) | 0 | 0 | 0.477465 |
| flags | GNB-CC | (0.443, 0.560, 0.428) | 0 | 0 | 0.473409 |
| flags | ML$k$NN | (0.307, 0.302, 0.233) | 0 | 0 | 0.278388 |
| flags | C2AE | (1.000, 1.000, 1.000) | 0 | 0 | 1.000000 |
| flags | CLIF | (0.298, 0.316, 0.217) | 0 | 0 | 0.273610 |
| flags | CLML | (0.281, 0.285, 0.205) | 0.025526 | 1.000000 | 0.254035 |
| medical | GNB-BR | (0.033, 0.576, 0.694) | 0 | 0 | 0.236867 |
| medical | GNB-CC | (0.030, 0.569, 0.686) | 0 | 0 | 0.226807 |
| medical | ML$k$NN | (0.018, 0.372, 0.284) | 0 | 0 | 0.124836 |
| medical | C2AE | (0.286, 0.858, 0.607) | 0 | 0 | 0.530094 |
| medical | CLIF | (0.013, 0.278, 0.134) | 0.175332 | 1.000000 | 0.079506 |
| medical | CLML | (0.028, 0.643, 0.496) | 0 | 0 | 0.207179 |

## A.5 EXTENDED RESULTS OF TRAINING CURVES AGAINST SURROGATE LOSS

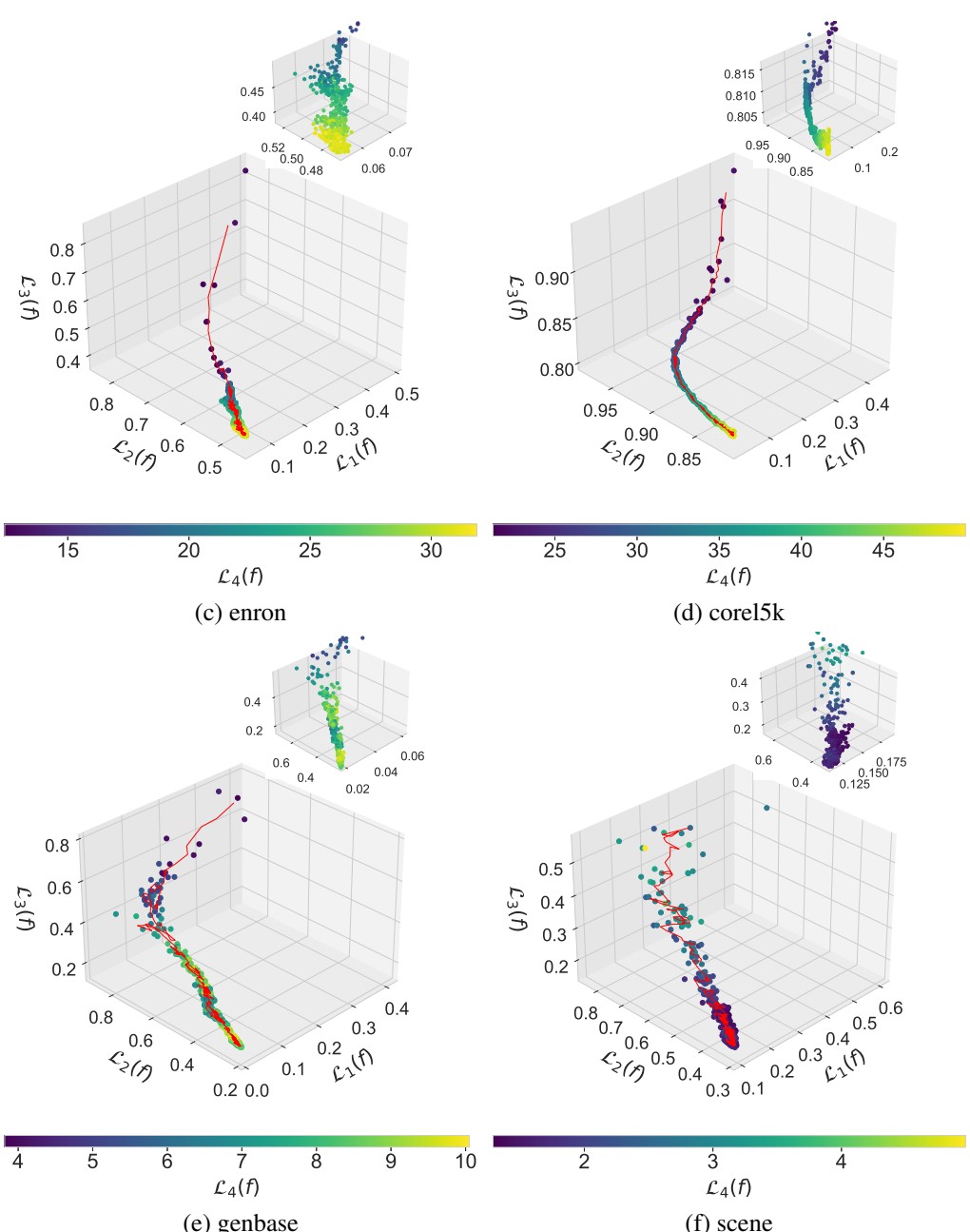

(c) enron

(d) corel5k

(e) genbase

(f) scene

Figure 7: The training curves of CLML on datasets enron through scene (c-f).

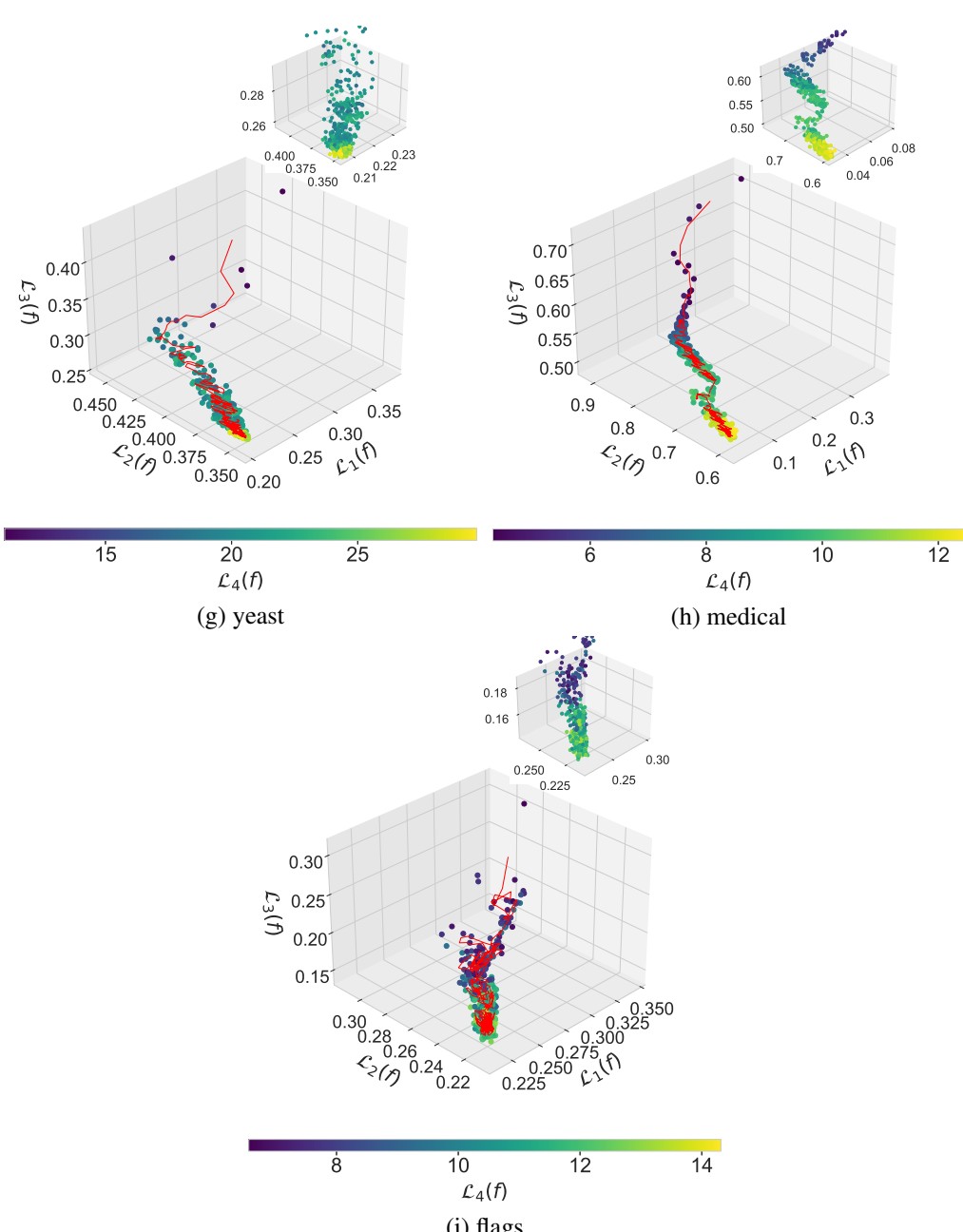

Figure 8: The training curves of CLML on datasets yeast through flags (g-i).

A.6    PROOF OF THEOREM 3.2

Here we show several useful and interesting Corollaries and Lemmas before proving Theorem 3.2.

**Definition A.1** (Metric Risk). *We define the conditional and Bayes risk of $\mathcal{L}_1, \mathcal{L}_2$, and $\mathcal{L}_3$ given $X$ and $Y$ for $i = 1, 2, 3$ as follows:*

$$R_{\mathcal{L}_i}(f) = \frac{1}{N} \sum_{j=1}^{N} \sum_{\boldsymbol{y}_j \in \mathcal{Y}} p(\boldsymbol{y}_j | \boldsymbol{x}_j) \mathcal{L}_i(f(\boldsymbol{x}_j), \boldsymbol{y}_j) \quad R_{\mathcal{L}_i}^B(f) = \frac{1}{N} \sum_{j=1}^{N} \inf_{f'}[\sum_{\boldsymbol{y}_j \in \mathcal{Y}} p(\boldsymbol{y}_j | \boldsymbol{x}_j) \mathcal{L}_i(f'(\boldsymbol{x}_j), \boldsymbol{y}_j)]$$

(19)

The overall risk and Bayes risk is given by:

$$R_{\boldsymbol{\mathcal{L}}}(f) = (R_{\mathcal{L}_1}(f), R_{\mathcal{L}_2}(f), R_{\mathcal{L}_3}(f)) \quad R_{\boldsymbol{\mathcal{L}}}^B(f) = (R_{\mathcal{L}_1}^B(f), R_{\mathcal{L}_2}^B(f), R_{\mathcal{L}_3}^B(f))$$

(20)

**Corollary A.0.1** (Below-bounded and Interval). *The Lebesgue measure is naturally below-bounded and interval, i.e., for any $F, F'$ and $R, R' \subset Z$, $\lambda(H(F, R)) = \lambda(H(F', R'))$ or $|\lambda(H(F, R)) - \lambda(H(F', R'))| > 0$, which is naturally inherited from the underlying below-bounded and interval properties of $\mathcal{L}_1, \mathcal{L}_2$ and $\mathcal{L}_3$ following Gao & Zhou (2011).*

**Lemma A.1** (The Lebesgue Contribution Equals Lebesgue Improvement). *Let $\lambda(H(F, R))$ denote the Lebesgue measure over a set $F$. The overall improvement toward the minimisation of $\mathcal{L}_1, \mathcal{L}_2$, and $\mathcal{L}_3$, is prescribed by the volume of $\lambda(H(F, R))$, which can be expressed as the sum of contributions of losses for each function representation $f \in F$:*

$$\lambda(H(F, R)) = \sum_{f \in F} \lambda(P(f)) = \sum_{f \in F} \int_{\mathbb{R}^o} \mathbf{1}_{H(\{f\}, R) \setminus H(F \setminus \{f\}, R)}(\boldsymbol{z}) d\boldsymbol{z}$$

(21)

*Proof.* Consider a redefined Lebesgue measure as the union of non-overlapping (disjoint) contribution regions for each $f \in F$. By substitution:

$$\lambda(H(F, R)) = \int_{\mathbb{R}^o} \mathbf{1}_{H(F, R)}(\mathbf{z}) d\mathbf{z} = \int_{\mathbb{R}^o} \mathbf{1}_{\cup_{f \in F} H(\{f\}, R) \setminus H(F \setminus \{f\}, R)}(\mathbf{z}) d\mathbf{z}$$

(22)

The integral can be re-written to express the sum over disjoint contribution regions:

$$\int_{\mathbb{R}^o} \mathbf{1}_{\cup_{f \in F} H(\{f\}, R) \setminus H(F \setminus \{f\}, R)}(\mathbf{z}) d\mathbf{z} = \sum_{f \in F} \int_{\mathbb{R}^o} \mathbf{1}_{H(\{f\}, R) \setminus H(F \setminus \{f\}, R)}(\mathbf{z}) d\mathbf{z} = \sum_{f \in F} \lambda(P(f)).$$

(23)
$\square$

*Proof of Theorem 3.2.* Recall the claim given a sequence $F^{(n)}$, the maximisation of the Lebesgue measure $\lambda(H(F^{(n)}, R))$ is consistent with the minimisation of $\mathcal{L}_1, \mathcal{L}_2$, and $\mathcal{L}_3$:

$$\lim_{n \to \infty} \lambda(H(F^{(n)}, R)) \to \lambda(H(\mathbb{P}^B, R)) \quad \text{then}$$
$$R_{\mathcal{L}_1}(f^{(n)}) \to R_{\mathcal{L}_1}^B(f) \wedge R_{\mathcal{L}_2}(f^{'(n)}) \to R_{\mathcal{L}_2}^B(f') \wedge R_{\mathcal{L}_3}(f^{''(n)}) \to R_{\mathcal{L}_3}^B(f'').$$

(24)

In other words, the maximisation of $\lambda(H(F^{(n)}, R))$ tends to the convergence toward the Bayes risk for each loss function $\mathcal{L}_i \ \forall i : 1 \le i \le 3$, $f^{(n)}, f^{(n)'}, f^{(n)''} \in F^{(n)}$ and that $f, f', f'' \in \mathbb{P}^B$.

We proceed by contradiction. Suppose the following function exists: $f^\gamma \notin \mathbb{P}^B$, $f^\gamma \in \Omega$ s.t. $\exists v : R_{\mathcal{L}_v}(f^\gamma) = R_{\mathcal{L}_v}^B(f^\gamma)$, i.e., $f^\gamma$ is a Bayes predictor for the $v^{th}$ loss $\mathcal{L}_v$ given $p(\mathbf{y}|\mathbf{x})$. Now suppose another function $f^\beta \in \Omega$ exists s.t. $f^\beta \in \mathbb{P}^B$. By this condition, $f^\beta \prec f^\gamma$ as $f^\gamma \notin \mathbb{P}^B$, hence $\forall i : 1 \le i \le o : \mathcal{L}_i(f^\beta) \le \mathcal{L}_i(f^\gamma)$, and $\exists \mathcal{L}_k : \mathcal{L}_k(f^\beta) < \mathcal{L}_k(f^\gamma)$. This result has two implications:

1. If $k = v$ then $\mathcal{L}_k(f^\beta) < \mathcal{L}_k(f^\gamma)$ would contradict $f^\gamma$ being a Bayes predictor. This would imply a Bayes predictor *cannot exist outside* $\mathbb{P}^B$.

2. If $k \neq v$, then $\forall i : 1 \leq i \leq o : \mathcal{L}_i(f^\beta) \leq \mathcal{L}_i(f^\gamma)$. For this condition to hold, when $i = v$, $\mathcal{L}_i(f^\beta) \leq \mathcal{L}_i(f^\gamma)$ would imply that $f^\beta$ is *also* a Bayes predictor of $\mathcal{L}_i$, when there is strict equality, and implication 1 when there is inequality. Therefore, the Bayes predictor of $\mathcal{L}_i$ already exists within $\mathbb{P}^B$.

The Pareto optimal set of representations therefore contains a set of Bayes predictors, one for each loss dimension. Hence, given a sequence $F^{(n)}$, the maximisation of the Lebesgue measure $\lambda(H(F^{(n)}, R))$ is eventually consistent with the minimisation of $\mathcal{L}_1, \mathcal{L}_2$, and $\mathcal{L}_3$.

$\square$

