# OpenReview forum: "A Consistent Lebesgue Measure for Multi-label Learning"
_ICLR.cc/2024/Conference — Submitted to ICLR 2024_

### Official Review · Reviewer_CLFN · 2023-10-29

**Soundness:** 2 fair
**Presentation:** 2 fair
**Contribution:** 2 fair
**Rating:** 6
**Confidence:** 2

**Summary:**

The paper aims to solve multi-label learning problem by jointly optimizing multiple objective function simultaneously. Considering that different loss functions may be potentially conflicting, the paper adopts a multi-objective optimization framework and solve the corresponding optimization problem by the traditional covariance matrix adaption (CMA-ES). Theoretical analysis on multi-label consistency is conducted on the proposed method.

**Strengths:**

1. The paper proposed a framework to optimize multiple loss functions simultaneously. Although this idea is borrowed from multi-objective learning, it is a key point to achieve good performance on multiple metrics (loss functions) in multi-label learning.

2. The paper proves the multi-label consistency for the proposed objective function although the main results are proved in the previous works.

**Weaknesses:**

1. I'm confused about one point: why did the author specifically focus on the learning model f? Does f have any impact on the proposed method? It seems to be a standard neural network. If there's anything I missed, please let me know.

2. The introduction of the proposed method Section 2.3 is unclear. The paper did not provide a detailed explanation of how to optimize the proposed objective function. If an existing method was used, the technical contribution of the proposed method is limited.

3. The experiments are weak due the following reasons:

1) The dataset is relatively small, with the largest dataset containing only 5000 examples.

2) The compared methods are not sufficiently advanced; the most recent method was proposed in 2022. Most of the other methods were introduced four to five years ago, or even earlier.

3) I think that the proposed method should be compared with some commonly used loss functions, such as binary cross entropy loss, ranking loss, etc, based on the same base model.

4) The figures were not plotted carefully; the font size of legends and axis is too small.

**Questions:**

How to optimize the objective function proposed in the paper?

---

> ### Author Response · Authors · 2023-11-13
> **Response to W1, W2, W3.1, W3.2, W3.3, and W3.4.**
>
> $\textbf{W1}$: Thank you for your question. We would like to make clarifications to help your understanding. The main focus of the paper is not the learning model $f$. The main focus of the paper is the consistent Lebesgue measure, hence, CLML is a model agnostic optimisation approach that could be applied to different classifiers. Although we use a standard neural network, CLML can be applied to other classifiers. We have also proven that the Lebesgue measure in CLML is consistent irrespective of the underlying classifier.
>
> $\textbf{W2}$: Thank you for your suggestion. This was brought up by previous Reviewers XYJ3 and Fh3C, and we have added additional technical exposition in the appendix of the updated manuscript. To be specific, have included a more detailed exposition of the underlying optimiser, and covariance matrix adaptation, in addition to an exposition on the Lebesgue measure and pseudocode that describes the overall optimisation procedure of CLML.
>
> $\textbf{W3.1}$: Thank you for your comment. In our experiments, we have chosen several datasets with over a thousand features, and several datasets with over 50 possible labels. In tabulated multi-label learning, the complexity of the problem can scale with the high possible number of interactions between features and labels, which we discuss in-depth in the appendix. Increasing the number of examples may not necessarily increase the complexity of the problem  (but might increase the experimentation time). In fact, comparing the smaller number of samples to the high degree of feature-to-label interaction dispersion across the dataset, it becomes quite challenging to train a classifier that can accurately predict a set of class labels.
>
> $\textbf{W3.2}$: We would like to draw attention to the lack of existing state-of-the-art methods in multi-label tabulated learning in recent years. To the best of our knowledge, CLIF represents the most state-of-the-art and recent multi-label tabulated learner.
>
> We have deliberately chosen CLIF as it is not only sufficiently advanced, but also outperforms the previously state-of-the-art methods that have been used extensively in existing research such as TIFS (2020), and MPVAE (2020). There are additional state-of-the-art and recent methods that focus on label distribution learning and computer vision, however, these are not applicable to the tabulated classification domain. CLML achieved a significant $13.4\%$ percent improvement over CLIF, and consistent with the findings of CLIF's respective paper, CLML also outperformed C2AE with a significant $55.5\%$ percent in improvement, on similar datasets and experimental setups. This implies that CLML is likely to also outperform previously state-of-the-art methods such as TIFS and MPVAE.
>
> $\textbf{W3.3}$: Thank you for your suggestion. We have already compared CLML against the widely-used cross-entropy loss (widely used as a surrogate loss function in most multi-label learning papers).
>
> Section 5 of the paper titled Effectiveness of the Lebesgue measure over surrogate loss is dedicated to this point. We also have several figures throughout the paper supporting this point. We have also deliberately chosen three of the most common multi-label loss functions: Hamming-loss, one minus Micro-$F_1$, and one minus label ranking average precision.
>
> $\textbf{W3.4}$: Thanks for your comment. We have increased the font sizes of several figures to improve readability.

---

> > ### Comment · Reviewer_CLFN · 2023-11-22
> > **Response to Rebuttal**
> >
> > Thank you very much for your great efforts on addressing my questions. I still have the following concerns:
> >
> > About SOTA results: Please refer to [1][2][3]. These methods could achieve comparable or better results than CLIF. I think this paper overclaimed its contributions according to "CLML can consistently achieve a better rank than state-of-the-art methods " in the abstract, since it was not compared with real SOTA.
> >
> > Novelty and Contribution: The main idea of this paper is very similar to [4], which also treated the multi-label learning task as a multi-objective optimization problem by considering multiple loss functions simultaneously. I find that the paper does not mentioned this work and discuss their differences. Due to the existence of this work, the innovation and contributions of the paper are overestimated. Moreover, the second point of motivation " how can a single model learn using multiple related, yet potentially conflicting loss functions?" also becomes invalid.
> >
> > [1] Disentangled variational autoencoder based multi-label classification with covariance-aware multivariate probit model
> > [2] End-to-End Probabilistic Label-Specific Feature Learning for Multi-Label Classification
> > [3] Dual Perspective of Label-Specific Feature Learning for Multi-Label Classification
> > [4] Multi-Label Classification Based on Multi-Objective Optimization

---

> ### Author Response · Authors · 2023-11-13
> **Response to question.**
>
> Thank you for your question. The optimisation is achieved using covariance matrix adaptation, wherein the quality of a solution is determined by its Lebesgue contribution, which is clearly described in Section 2.3. Additional technical exposition has been included in the Appendix of the updated manuscript to further elaborate on this, including a description of covariance matrix adaptation, estimation of the Lebesgue measure, and pseudocode of the proposed CLML.

---

> ### Author Response · Authors · 2023-11-22
> **Reviewer CLFN additional concerns**
>
> Thank you for bringing [4] to our attention. First, we would like to address the novelty and contribution concerns. The paper in question addresses multi-objective optimisation using a dominance-based sorting mechanism. The major contributions of our work, and what ultimately sets it apart from the aforementioned paper, is that we aim to: (1) consistently optimise a set of non-differentiable or non-convex multi-label loss functions, where consistency is defined under the Bayes risk framework, and (2) propose to use the Lebesgue measure to achieving such an optimisation, which can be expressed analytically.
>
> The significance of our paper is that we first $\textit{prove}$ that the Lebesgue measure is capable of achieving such consistency. The paper you have mentioned does not prove that dominance-based sorting is capable of achieving consistency (although they briefly mention consistency as a motivation, citing Gao 2011). This sets us apart from the aforementioned paper in that we have a significant theoretical contribution. Second, regarding the analytical expression of the Lebesgue measure, this opens up future research avenues to further build upon CLML. Mainly, we envision that CLML can be applied to contemporary problems such as multi-label image classification (with millions of model parameters) by modifying the underlying Lebesgue measure to achieve differentiation-based optimisation (by approximating certain parts of the Lebesgue estimation for automatic differentiation). The aforementioned paper is GA-based, and cannot realistically be applied to such contemporary problems, mainly since dominance-based sorting cannot be expressed analytically, and GA-based search algorithms are inefficient and ineffective at optimising neural network model parameters. We argue that this paper offers a significant step toward a consistent optimisation process for all multi-label problems, although fully achieving this in one paper falls outside of scope.
>
> Thank you for pointing out the additional papers regarding the SOTA concerns. The first paper Disentangled variational autoencoder based multi-label classification with covariance-aware multivariate probit model [1] is compared against CLIF in its original corresponding paper. CLIF achieves better performance than MP-VAE in most cases. Regarding End-to-End Probabilistic Label-Specific Feature Learning for Multi-Label Classification [2] and Dual Perspective of Label-Specific Feature Learning for Multi-Label Classification [3], our investigation of state-of-the-art methods led to many papers that employed label graph embedding and semantic embedding techniques, which was suggestive to compare against state-of-the-art methods that performed label graph embedding and semantic embedding, ultimately leading us to CLIF. Although we cannot make these comparisons now (before the end of the discussion period), we thank you for bringing this to our attention. However, we would like to further urge that our paper still maintains a clear and significant theoretical contribution in that consistent optimisation can be achieved in the context of optimising multiple loss functions. As we discussed in our previous point, CLML paves the way for future research to reconsider the optimisation process of multi-label learners to achieve theoretical consistency. In fact, the papers that you have pointed out still use cross-entropy-based loss functions to estimate multi-label classification performance. We hope that our paper can entice researchers to reconsider the optimisation of such advanced and state-of-the-art methods to use a consistent measure such as the Lebesgue measure proposed in our paper to drive performance ever closer toward a theoretical upper limit.

---

> > ### Comment · Reviewer_CLFN · 2023-11-23
> > **Response to Rebuttal**
> >
> > Thank you very much for your feedback.

---

### Official Review · Reviewer_Fh3C · 2023-10-30

**Soundness:** 3 good
**Presentation:** 2 fair
**Contribution:** 3 good
**Rating:** 6
**Confidence:** 4

**Summary:**

The article introduces the challenges associated with multi-label loss functions, particularly their non-convex or discontinuous nature, which makes direct optimization problematic. Recognizing the inconsistencies between surrogate loss functions and their desired counterparts, the authors present a novel approach termed the "Consistent Lebesgue Measure-based Multi-label Learner" (CLML). This technique posits that optimizing the Lebesgue measure directly correlates to the optimization of various multi-label losses. Under a Bayes risk framework, the authors demonstrate the theoretical consistency of CLML.

**Strengths:**

1.	The paper introduces a novel learning objective adept at handling various non-convex and discontinuous multi-label loss functions, which notably eliminates the need for surrogate loss functions. This innovative approach offers a more direct and potentially efficient solution to challenges associated with traditional multi-label loss optimization.
2.	The authors have provided clear proof showing that their method is consistent. This adds credibility to their approach and assures users of its reliability in various applications.
3.	The authors have conducted extensive experiments to demonstrate the advantages of their method. This thorough empirical validation underscores the effectiveness and robustness of their approach in real-world scenarios.

**Weaknesses:**

1. There appears to be a typographical error in Equation 2. Specifically, in the loss function L\left(f\left(x\right),y\right), the y should likely be denoted as y^\prime. This inconsistency needs to be rectified to ensure clarity and coherence in the formulation.
2. The optimization method in the last paragraph of section 2.3 is not elaborated in detail. To enhance clarity and provide readers with a holistic understanding, the authors are recommended to furnish a more detailed exposition on the employed optimization techniques.
3. The methodology proposed in this paper appears to be tailored specifically for tabular data. This inherent design might limit its applicability to large-scale image datasets, which inherently possess a different data structure and complexity. Thus, the generalizability of the approach to diverse data types, especially image datasets, remains questionable.

**Questions:**

1. In Equation 2, can the authors clarify the notation used in the loss function L\left(f\left(x\right),y\right)? Is there a specific reason for using y instead of the seemingly more appropriate y^\prime?
2. Regarding the optimization method presented in the latter part of section 2.3, could the authors delve deeper into the specifics of this method?
3. Given that the paper's methodology seems predominantly designed for tabular data, how do the authors envision adapting or evolving this method for more complex datasets, such as large-scale image datasets?
4. Have the authors considered expanding the variety of multi-label loss functions to further enhance the model's performance? Additionally, would assigning different weights to distinct loss functions potentially lead to further improvements in the model's efficacy? It would be interesting to understand the impact of such variations on the overall performance.

---

> ### Author Response · Authors · 2023-11-13
> **Response to W1, W2, and W3.**
>
> $\textbf{W1}$: Thank you for pointing this out. Reviewers PXNK and XYJ3 have also previously pointed this out. We have corrected this minor typographical error in the updated manuscript.
>
> $\textbf{W2}$:  Thank you for the valid point. Reviewer XYJ3 also shared the same concern. We have included some additional formulas and explanations in the appendix to clarify the optimisation method, specifically regarding Section 2.3. To be specific, have included a more detailed exposition of the underlying optimiser, covariance matrix adaptation, in addition to an exposition on the Lebesgue measure and pseudocode that describes the overall optimisation procedure of CLML.
>
> $\textbf{W3}$: Thank you for your comment. As you have shared the same concern as reviewer XYJ3, we will elaborate on a similar point. Although many of the state-of-the-art multi-label papers focus on multi-label object recognition, tabulated data remains an important research area in multi-label learning.
>
> Furthermore, the purpose of this paper was to present concrete theoretical and empirical evidence that consistency can be achieved using the Lebesgue measure. Therefore, CLML and the contributions of this paper are not intended to be limited to tabulated data but instead are supposed to focus on communicating an effective alternative to traditional optimisation methods for multi-label learning.
>
> We are hoping that our paper will pioneer future research in multi-label learning by enticing researchers to reconsider the optimisation procedure in multi-label learning. Understandably, applying CLML to contemporary problems efficiently, such as large-scale image datasets, requires additional work due to the additional complexity of the data and learners, which we felt would detract the main focus of this paper from demonstrating the effectiveness of CLML in the aforementioned theoretical and empirical senses. Incorporating computer vision would have significantly increased the length (or necessary content) of the paper and would have potentially fallen out of scope. Again, we would like to stress that the intention of this paper is to provide grounds for an alternative optimisation approach that future work can build upon for more complex tasks.
>
> We would like to continue elaborating on this point as we address your questions in the following discussions.

---

> ### Author Response · Authors · 2023-11-13
> **Response to Q1, Q2, Q3, and Q4.**
>
> $\textbf{Q1}$: Thank you for pointing this out. The other reviewers have also commented on this. We have corrected this minor typographical error in the updated manuscript.
>
> $\textbf{Q2}$: Thank you for your question. We have included some additional technical exposition, formulas, and explanations in the appendix to clarify the optimisation in Section 2.3. We have included a more detailed exposition of the underlying optimiser, covariance matrix adaptation, in addition to an exposition on the Lebesgue measure and pseudocode that describes the overall optimisation procedure of CLML.
>
> $\textbf{Q3}$: This is indeed an interesting problem and the future direction will require a lot of work. Contemporary machine-learning tasks related to large-scale image datasets utilise models with much higher complexity than the standard neural network used in this paper.
>
> We would kindly ask the reviewer to also see our response to Reviewer XYJ3 who shared a similar question. Currently, CLML can be applied directly to image classification problems with only minor amendments to the implementation. However, due to our computational constraints, we chose to focus this work on developing and proving a consistent optimisation method to work on tabulated data. Adapting CLML to contemporary problems $\textit{efficiently}$ like computer vision requires additional research (which we thought would fall out of scope from the main contributions of this paper), and may require finding alternate ways to differentiate or design a differentiable approximate of the Lebesgue measure to enable migrating back to gradient-based search. Naturally, this would also require a thorough investigation on whether such an approximate would still be consistent with the desired loss functions.
>
> $\textbf{Q4}$: Thank you for your questions. Regarding the first point, we have used three of the most commonly used multi-label loss functions. Increasing the variety of loss functions will not necessarily enhance model performance, $\textit{i.e.}$, loss functions that are not relevant to the specific multi-label learning problem may not necessarily guide the learner toward promising solutions. There may also be some overlap between the information quantified between loss functions, therefore increasing the sheer number of loss functions optimised after a certain point may reflect diminishing returns.
>
> We chose Hamming-Loss, Micro-$F_1$, and label ranking average precision deliberately knowing that: (1) we wanted to improve the performance of each loss function specifically; and (2) there exists some potential conflict between them, so it is necessary to optimise them jointly if we are interested in finding solutions to the trade-offs.
>
> Regarding your second point, weighting the loss functions is not required under the Lebesgue measure, and can be potentially detrimental to the performance. Weighing the distinct loss functions to determine the relative importance may not guarantee consistency. Weight vectors may direct the optimiser away from the Bayes risk of a subset of the loss
> functions if those loss functions are weighted low. For a fair comparison, weighting methods require careful design of weight vector distribution which over-complicates the matter and
> falls far outside the scope of this paper. Furthermore, the Lebesgue measure naturally accounts for the relations between distinct loss functions, as it is an analytical method that integrates over the area covered by a loss vector in a multi-dimensional loss landscape. We kindly refer the reader to see evidence in the training loss curves in Fig. 7 (d) and (e) in the appendix. The training curves demonstrate that earlier during the optimisation process, $\mathcal{L}_1$ and $\mathcal{L}_3$ are minimised more effectively than $\mathcal{L}_2$. Eventually, CLML is able to correct itself to further improve $\mathcal{L}_2$. These patterns indicate a complex loss landscape with varying difficulty in each loss dimension that CLML can automatically account for. Knowing this behaviour beforehand may suggest that $\mathcal{L}_2$ needs to be weighted differently than $\mathcal{L}_1$ and $\mathcal{L}_3$, although eventually CLML can cope with these differences without such methods.

---

> > ### Comment · Reviewer_Fh3C · 2023-11-21
> >
> > I appreciate the author's response, which addresses most of my concerns. Having reviewed the author's response and the comments from other reviewers, I decide to maintain the score.

---

### Official Review · Reviewer_XYJ3 · 2023-10-31

**Soundness:** 3 good
**Presentation:** 2 fair
**Contribution:** 3 good
**Rating:** 6
**Confidence:** 4

**Summary:**

The article introduces the challenges associated with multi-label loss functions, particularly their non-convex or discontinuous nature, which makes direct optimization problematic. Recognizing the inconsistencies between surrogate loss functions and their desired counterparts, the authors present a novel approach termed the "Consistent Lebesgue Measure-based Multi-label Learner" (CLML). This technique posits that optimizing the Lebesgue measure directly correlates to the optimization of various multi-label losses. Under a Bayes risk framework, the authors demonstrate the theoretical consistency of CLML.

**Strengths:**

1. The introduction of a groundbreaking learning objective that adeptly addresses non-convex and discontinuous multi-label loss functions, significantly eliminating the dependence on surrogate loss functions, showcases a more streamlined and potentially more efficient approach to traditional multi-label loss optimization challenges.
2. The clear proof of method consistency provided by the authors lends strong credibility to their novel approach, ensuring its reliability and applicability across diverse scenarios.

**Weaknesses:**

1. A potential weakness of the algorithm is its limitation to tabular data, which may render it ineffective for handling large-scale image datasets prevalent in contemporary research and applications.
1. In Section 2.3, the optimization algorithms are merely summarized in brief statements rather than being detailed through comprehensive formulas. It raises the question of whether the complexity of these optimization methods contributes to the model's inability to handle large-scale datasets.
1. In Equation (2), the loss function and surrogate loss function incorrectly use $ y $ instead of $ y' $. This appears to be a typographical error.

**Questions:**

Could the authors provide more detailed formulations or explanations for the optimization methods mentioned in Section 2.3? I'm curious to understand their specifics. Additionally, could the authors clarify whether these methods are scalable to handle large-scale datasets?

---

> ### Author Response · Authors · 2023-11-13
> **Response to W1, W2, and W3.**
>
> $\textbf{W1}$: Thank you for your comment. Although many of the state-of-the-art multi-label papers focus on multi-label object/image recognition, tabulated data remains an important research area in multi-label learning.
>
> The purpose of this paper was to present concrete theoretical and empirical evidence that consistency can be achieved using the Lebesgue measure. Therefore, CLML and the contributions of this paper are not intended to be limited to tabulated data (in fact, CLML can technically be applied to image classification problems, and in-fact most domains including regression and combinatorial optimisation, with only minor amendments to the code), but instead are supposed to focus on communicating an effective alternative to traditional optimisation methods for multi-label learning. We would kindly urge the reviewer to read our response to Reviewer Fh3C's W3. and Q3. (both of which relate to the topic of computer vision), where we provide further elaboration in the context of future work.
>
> From here, we are hoping that our paper will pioneer future research in multi-label learning by enticing researchers to reconsider the optimisation procedure in multi-label learning. Although CLML can be applied to contemporary problems such as large-scale image datasets, much additional work is required to address efficiency due to the additional complexity of the data and learners, which we felt would detract the main focus of this paper from demonstrating the effectiveness of CLML in the aforementioned theoretical and empirical senses. Incorporating computer vision efficiently and effectively would have significantly increased the length (or necessary content) of the paper and would have potentially fallen out of scope. Again, we would like to stress that the intention of this paper is to provide grounds for an alternative optimisation approach that future work can build upon for more complex tasks.
>
> $\textbf{W2}$: Thank you for the valid point. We have included some additional formulas and explanations in the appendix to clarify the optimisation, specifically regarding Section 2.3. We have included a more detailed exposition of the underlying optimiser, covariance matrix adaptation, in addition to exposition on the Lebesgue measure and pseudocode that describes the overall optimisation procedure of CLML.
>
> $\textbf{W3}$: Thank you for pointing this out. Reviewer PXNK and Fh3C also commented on this. We have corrected this minor typographical error in the updated manuscript.

---

> ### Author Response · Authors · 2023-11-13
> **Response to Questions.**
>
> Thank you for the question! We have included a more detailed exposition of the underlying optimiser, covariance matrix adaptation, in addition to an exposition on the Lebesgue measure and pseudocode that describes the overall optimisation procedure of CLML. We hope that our revision will clarify the optimisation methods in Section 2.3 to a satisfactory degree.
>
> In tabulated multi-label learning, the complexity of the learning problem can scale with the high number of interactions between features and labels, which can be a challenge even for datasets with a relatively small number of examples. We discuss this point in-depth in the appendix. The computational complexity of CLML during inference scales linearly with the number of features and labels. Therefore, a majority of the current complexity arises from the covariance update step, which is quadratic with the number of parameters in the network. Due to our computational constraints, we have chosen to focus this work on proving that consistent optimisation in multi-label learning can be achieved, while presenting empirical evidence to support our theory, on tabulated learning problems. However, this does not limit our work to tabulated problems, in fact, we have proven the consistency of CLML irrespective of the underlying classifier/task. CLML can be adapted to contemporary problems such as computer vision with minor amendments to the code. The challenge is that doing so efficiently would require some additional research (which we thought would fall out of scope from the main contributions of this paper). For this reason, we chose to focus on demonstrating the empirical and theoretical advantages of CLML against state-of-the-art multi-label learners on tabulated problems to entice further research in multi-label learning to reconsider the optimisation process.

---

> > ### Comment · Reviewer_XYJ3 · 2023-11-22
> >
> > Thank you for the response, which addressed most of my concerns. I have decided to maintain my original score.

---

### Official Review · Reviewer_PXNK · 2023-11-05

**Soundness:** 2 fair
**Presentation:** 2 fair
**Contribution:** 2 fair
**Rating:** 3
**Confidence:** 2

**Summary:**

This paper aims to leverage the existing Lebesgue measure to propose a method capable of comprehensively considering a range of multi-label loss functions to ensure their effectiveness. The paper also conducts experiments to validate the performance of this method with different loss functions on several datasets.

**Strengths:**

1. This paper touches on a crucial issue in multi-label learning: different loss functions measure classifier performance differently.

2. The proposed method in this paper is based on neural networks, allowing it to benefit from the advancements in neural network technology.

**Weaknesses:**

1. The motivation behind this paper is somewhat weird. Given the knowledge that different loss functions lead to different outcomes, why introduce a "universal method" instead of determining which loss function to use based on practical needs?

2. This paper is an application of the Lebesgue method, and there are numerous other methods for multi-objective optimization. The paper does not compare its method with other multi-objective optimization techniques.

3. The technical exposition in this paper is unclear and contains several issues.

4. The method proposed in this paper does not exhibit a significant empirical difference from the latest method, CLIF. The experimental results are rather marginal improvement.

**Questions:**

1. What is the specific rationale behind choosing Lebesgue's multi-objective optimization method?

2. In cases where classifiers resulting from different loss functions may conflict, why is the universal method proposed in this paper still necessary?

3. Is there an issue with defining p(x) as equivalent to p(x|y) (above Eq 1), where the former depends solely on x, while the latter depends on both x and y simultaneously?

4. Is there a problem with Eq 2, where the latter y should be y'?

5. In the absence of a specified R, how does optimizing lambda(P(f)) achieve multi-objective optimization? If R consists of only one element, it is possible that H(F, R) could be empty.

---

> ### Author Response · Authors · 2023-11-13
> **Response to W1, W2, W3, and W4**
>
> $\textbf{W1}$: Thank you for your comment, this raises a good point. We acknowledge the importance of tailoring approaches to specific requirements, however, our argument for a universal method stems from the potential for certain problems to possess multiple related and relevant loss functions.
>
> First, CLML represents a universal method that streamlines the optimisation process to achieve consistent optimisation w.r.t. the desired loss function(s). Suppose you know that $\mathcal{L}_1$ and $\mathcal{L}_2$ are related to your multi-label classification task and that you care about both (and that you do not care about $\mathcal{L}_3$). Traditional gradient-based methods still require a manually designed and differentiable objective function to surrogate $\mathcal{L}_1$ and $\mathcal{L}_2$. On the other hand, CLML can optimise $\mathcal{L}_1$ and $\mathcal{L}_2$ directly. The same can apply to any permutation of the set of applicable loss functions that may relate to the problem. Another example is if $\mathcal{L}_1$ is the only relevant loss function to the problem. CLML can certainly optimise $\mathcal{L}_1$ in isolation (ignoring both $\mathcal{L}_2$ and $\mathcal{L}_3$). The key contribution here is that CLML can not only consider the desired loss function directly but also the interaction and conflict between multiple loss functions in the event that multiple are relevant to the target problem.
>
> We argue that this does not dismiss the importance of determining which loss functions are important based on practical needs. Instead, CLML serves as a versatile tool that can adapt to different practical needs (ones involving one or more related loss functions), allowing for a more holistic, consistent, and automated approach to optimisation for multi-label learning.
>
> $\textbf{W2}$: Thanks again for your suggestion. First, we would like to point out that although there is a lot of research on multi-objective optimisation, there is scarce work on multi-objective classification for multi-label data (even more so for tabulated data). To the best of our knowledge, this area has not been explored sufficiently in our specific problem domain. Furthermore, most existing multi-objective optimisation methods cannot directly be applied to multi-label learning problems, and instead are suited to single-label (binary or multi-class) classification, regression, or even combinatory optimisation problems. The major contributions of our work, and what ultimately sets it apart from other multi-objective papers, is that we aim to: (1) consistently optimise a set of non-differentiable or non-convex multi-label loss functions, where consistency is defined under the Bayes risk framework, and (2) propose to use the Lebesgue measure to achieving such an optimisation, which can be expressed analytically. Expressing the Lebesgue measure analytically is paramount as it opens up future research to approximate certain parts of the Lebesgue measure to suit automatic differentiation. This may enable consistent gradient-based optimisation.
>
> Regarding the specific use of the Lebesgue measure as a quantifier, there are certainly other methods for quantifying performance in a multi-dimensional loss landscape. However, the key contribution was to achieve consistent optimisation over the desired loss functions using the Lebesgue measure. This is not always the case for other multi-objective measures.
>
> For example, decomposition-based methods can use the Tchebycheff aggregation of a set of weighted loss functions. The consistency of decomposition-based methods can not be guaranteed since the weight vectors may direct the optimiser away from the Bayes risk of a subset of the loss functions. For a fair comparison, decomposition-based methods require careful design of weight vector distribution, which over-complicates the matter and falls far outside the scope of this paper.
>
> $\textbf{W3}$: Thanks for your comments. We have addressed all of your points. We have also expanded on the technical exposition of the paper related to Section 2.3 in the appendix. We have included a more detailed exposition of the underlying optimiser, covariance matrix adaptation, in addition to an exposition on the Lebesgue measure and pseudocode that describes the overall optimisation procedure of CLML.
>
> $\textbf{W4}$: Thanks for your comment. We would like to emphasise that CLML achieved a $13.4$ percent overall improvement in performance against CLIF. This represents a clear improvement over a state-of-the-art method that previously outperformed many well-known and state-of-the-art multi-label learners.
>
> The improvement was achieved with a simpler neural network that did not utilise $\textit{any}$ of the state-of-the-art label graphs or semantic embeddings that CLIF uses. Furthermore, the results support our theory signifying the importance of consistency in achieving an overall better learning performance.

---

> ### Author Response · Authors · 2023-11-13
> **Response to Q1, Q2, Q3, Q4, and Q5.**
>
> $\textbf{Q1}$: Thank you, that is a good question. We would like to refer the reviewer to our answer to W.2 for some additional insight, which we will also further elaborate on here to answer your question.
>
> One of our main contributions was to consistently optimise a set of non-differentiable or non-convex multi-label loss functions, where consistency is defined under the Bayes risk framework. Without loss of generality, the set can consist of any number of multi-label loss functions that may or may not be practical to the target domain. The Lebesgue measure integrates over an area between a reference point (which we define as the origin vector in multi-dimensional space) and a multi-dimensional point that corresponds to a loss vector generated from some arbitrary model $f$.
>
> The Lebesgue measure has some desirable properties which make it a natural choice for optimising multiple loss functions simultaneously. First, the area covered by a loss vector is determined simply by its position in the multi-dimensional loss landscape, where the position in each dimension corresponds to a value obtained by a desired loss function. This property ensures that an improvement of the Lebesgue measure directly, and analytically, corresponds to an improvement of the underlying desired loss functions. Second, in comparison to other multi-objective methods which were covered in our response to W.2, the Lebesgue measure does not require specifying weight vectors for each loss function, $\textit{i.e.}$, the relative importance between loss functions do not need to be known prior to the optimisation process. This makes it quite effective to apply the Lebesgue measure on novel problems that have no prior domain knowledge. To support this, we kindly refer the reviewer to the training curves in Fig. 7 (d) and (e) in the appendix. The training curves demonstrate that earlier during the optimisation process, $\mathcal{L}_1$ and $\mathcal{L}_3$ are minimised more effectively than $\mathcal{L}_2$. Eventually, CLML is able to correct the trajectory to further improve $\mathcal{L}_2$. These patterns indicate a complex loss landscape with varying difficulty in each loss dimension that CLML can automatically account for.
>
> $\textbf{Q2}$: Thanks for your question, this is a good point. We have partially addressed this in W1., and we will elaborate on this topic further here.
>
> It is still essential to propose a universal method. There are two scenarios to consider, both of which can be addressed by CLML. First, CLML can find a classifier that obtains an overall good performance with trade-offs on several loss functions. This can be desirable in problems where there is a set of applicable or practical loss functions that need to be considered in tandem. In many real-world problems, it is essential to find a solution that can achieve acceptable performance on several metrics, and not a single metric at the expense of others. Second, according to our proof, we also demonstrate that CLML can eventually find a Bayes risk predictor for each of the underlying desired loss functions as it is consistent with each loss function (assuming that one abstracts away the classifier). During optimisation, these Bayes risk predictors can be archived. Hence, CLML can potentially find optimal classifiers for each function $\textit{and}$ find good classifiers with desirable trade-offs between all loss functions.
>
> $\textbf{Q3}$: Thank you for bringing this to our attention. We originally denoted $p(\textbf{x})$ as the conditional probability $p(\textbf{y}|\textbf{x})$ for simplicity and stated so in the paragraph above Eq. 1. However, in retrospect this can cause unnecessary confusion so we have rectified this by clearly stating the full conditional probability $p(\textbf{y}|\textbf{x})$ in the updated manuscript.
>
> $\textbf{Q4}$: Thank you for pointing this out. Reviewers XYJ3 and Fh3C also commented on this. We have corrected this minor typographical error in the updated manuscript.
>
> $\textbf{Q5}$: Thank you for your question. This is a good point. Initially, $R$ consists of the unit vector in the multi-dimensional loss landscape. We have included this information in the updated manuscript. The unit vector denotes the worst possible performance on all loss functions. From there, CLML will find a function that can achieve better performance than the unit loss vector. During the early stages of the optimisation it is likely that the set of mutually-non dominating vectors will be small, which runs the risk of $H(F,R)$ being empty. However, we use the Monte Carlo sampling method to sample loss vectors in $Z$ to approximate the actual hypervolume efficiently. This mitigates the risk of $H(F,R)$ being empty. This point may become more clear with the additional exposition that we have provided in the abstract relating to the optimisation procedure in Section 2.3.

---

> ### Author Response · Authors · 2023-11-23
> **Reviewer PXNK remarks**
>
> Dear reviewer,
>
> As the deadline for the author-reviewer discussion approaches, we would like to touch base and ask if the reviewer has any additional comments or concerns that we have not yet addressed.
>
> Kind regards

---

### Author Response · Authors · 2023-11-21
**Revisions based on reviewer comments**

Dear reviewers,

Thank you very much for taking the time to give us detailed feedback regarding the initial manuscript. We would like to kindly remind you that we have addressed all of your feedback and questions, and have included an updated manuscript for you to consider.

All the best,

Authors of submission 2175

---

### Meta-Review · Area_Chair_ptZY · 2023-12-08

**Metareview:**

This paper targets the improvement of existing multi-label loss functions, receiving ratings of 6, 6, 6, and 3.

During the rebuttal, reviewers acknowledged that: 1) it contributes to an important problem; 2) theoretical explanations are provided.

However, during the internal discussion, no reviewers championed the paper. Reviewers acknowledge that some concerns have not been well addressed.

Specifically, Reviewer Fh3C, despite a positive rating, mentioned that after reading all rebuttals and comments, he/she decided not to champion the paper for acceptance.

Reviewer XYJ3 and Reviewer CLFN, despite positive ratings, also acknowledged the limitations in practical usefulness and originality.

Additionally. some concerns regarding originality, insights, and evaluation have also been raised by the reviewers during the internal discussion.

- Clearly explain the unique change in the setting of multi-objective classification for multi-label data compared to multi-objective classification for single-label data.
- A more comprehensive evaluation of the method on different scales of datasets is required to convince readers. Currently, a dataset with a sample size of 5000 is not convincing.

**Justification For Why Not Higher Score:**

Overall, this is an interesting paper. However, concerns have been raised about the advantages of their method. For instance, it is unclear why existing multi-objective classification methods cannot be simply adapted to apply to the multi-label setting. In the rebuttal, only the conclusion was provided without a convincing explanation. Additionally, the evaluation of the method has not been convincingly presented. We believe that the paper could be significantly strengthened by addressing these concerns in the next round.

**Justification For Why Not Lower Score:**

NA

---

### Decision · Program_Chairs · 2024-01-16

Reject